# LiteGuard: Efficient Task-Agnostic Model Fingerprinting with Enhanced Generalization

**Guang Yang**[1]**, Ziye Geng**[2]**, Yihang Chen**[2]**, and Changqing Luo**[2]

[1]Virginia Commonwealth University, [2]University of Houston

yangg2@vcu.edu, {zgeng2, ychen165, cluo3}@uh.edu

## Abstract

Task-agnostic model fingerprinting has recently gained increasing attention due to its ability to provide a universal framework applicable across diverse model architectures and tasks. The current state-of-the-art method, MetaV, ensures generalization by jointly training a set of fingerprints and a neural-network-based global verifier using two large and diverse model sets: one composed of pirated models (i.e., the protected model and its variants) and the other comprising independently-trained models. However, publicly available models are scarce in many real-world domains, and constructing such model sets requires intensive training efforts and massive computational resources, posing a significant barrier to practical deployment. Reducing the number of models can alleviate the overhead, but increases the risk of overfitting, a problem further exacerbated by MetaV's entangled design, in which all fingerprints and the global verifier are jointly trained. This overfitting issue leads to compromised generalization capability of verifying unseen models.

In this paper, we propose LiteGuard, an efficient task-agnostic fingerprinting framework that attains enhanced generalization while significantly lowering computational cost. Specifically, LiteGuard introduces two key innovations: (i) a checkpoint-based model set augmentation strategy that enriches model diversity by leveraging intermediate model snapshots captured during the training of each pirated and independently-trained model—thereby alleviating the need to train a large number of pirated and independently-trained models, and (ii) a local verifier architecture that pairs each fingerprint with a lightweight local verifier, thereby reducing parameter entanglement and mitigating overfitting. Extensive experiments across five representative tasks show that LiteGuard consistently outperforms MetaV in both generalization performance and computational efficiency.

## 1 Introduction

Model fingerprinting has been considered a promising technique for safeguarding the ownership of deep neural networks (DNNs) (Chen et al., 2022; Xu et al., 2024; Godinot et al., 2025; Zhang et al., 2025). As valuable digital assets, DNN models often become prime targets for adversaries seeking unauthorized use and redistribution (Chen et al., 2021; Waheed et al., 2024; Stang et al., 2024; Yao et al., 2025a). For instance, adversaries may steal DNN models and publicly deploy these pirated models as a service. To protect the intellectual property (IP) of such DNN models, model fingerprinting exploits a model's inherent characteristics to generate fingerprints during the fingerprint generation stage, and utilizes these fingerprints to verify the ownership of suspect models during the fingerprint verification stage. However, adversaries may intentionally evade ownership verification by exploiting ownership obfuscation techniques (Pan et al., 2022; Zhao et al., 2024; Xu et al., 2024; Yao et al., 2025b) to modify the stolen models without degrading their utility, and/or publicly deploying these models as cloud services. To counter such threats, model fingerprinting typically crafts inputs that elicit distinctive model outputs, and uses the resulting input-output pairs as unique fingerprints to verify model ownership.

Researchers have developed model fingerprinting methods that can be broadly categorized into task-specific and task-agnostic approaches. To the best of our knowledge, most existing methods are task-specific, with a particular focus on classification tasks (Zhao et al., 2020; Cao et al., 2021;

Wang et al., 2021b; Yin et al., 2022; Yang & Lai, 2023; Liu & Zhong, 2024; Lukas et al., 2021; Li et al., 2021; Ren et al., 2023; Peng et al., 2022; Guan et al., 2022; Xu et al., 2024; Godinot et al., 2025; Tang et al., 2025). For example, many of them leverage adversarial examples to craft fingerprints that reflect the unique characteristics of a model's decision boundaries (Zhao et al., 2020; Cao et al., 2021; Wang et al., 2021b; Yin et al., 2022; Yang & Lai, 2023; Liu & Zhong, 2024; Xu et al., 2024; Godinot et al., 2025), thereby effectively distinguishing the protected model and its variants from independently-trained ones. Some works consider evading verification through ownership obfuscation techniques, such as fine-tuning and pruning (Lukas et al., 2021; Li et al., 2021; Ren et al., 2023). Beyond the classification task, a few recent studies have begun exploring fingerprinting for non-classification tasks. For instance, You et al. (2024); Waheed et al. (2024) generates fingerprints based on node features and graph topology to protect the ownership of graph neural networks (GNNs). Similarly, other researchers propose training classifiers to learn distinct fingerprints from images produced by Generative Adversarial Networks (GANs) to protect their ownership (Yu et al., 2019; Huang et al., 2023). While these task-specific fingerprinting methods have shown effectiveness in ownership protection, their reliance on task-specific characteristics inherently restricts their applicability beyond their target tasks.

In contrast, task-agnostic fingerprinting aims to provide broad applicability across diverse model architectures and tasks. To date, two task-agnostic approaches have been proposed: TAFA (Pan et al., 2021) and MetaV (Yang et al., 2022). However, TAFA assumes ReLU activations and continuous model outputs, restricting its practical applicability and preventing it from being fully task-agnostic. MetaV, by contrast, imposes no architectural assumptions and remains the only fully task-agnostic approach. It adopts an end-to-end framework that jointly trains a set of fingerprints and a neural-network-based global verifier. The training utilizes a piracy set composed of a protected model and its variants and an independence set comprising independently-trained models. During ownership verification, all fingerprints are passed through a suspect model, and the resulting outputs are concatenated and fed into the global verifier to produce a confidence score indicating the likelihood of the suspect model being a pirated copy of the protected one.

However, MetaV ensures generalization by heavily relying on access to large and diverse model sets during fingerprint training. In practice, publicly available models are scarce in many real-world domains, and constructing large model sets incurs intensive training efforts and prohibitive computational costs. While reducing the size of the model sets can alleviate computational burden, it significantly increases the risk of overfitting, i.e., the jointly trained fingerprints and global verifier tend to overfit to the limited training models, consequently resulting in poor generalization to unseen models. This issue is further exacerbated by the entangled training of all the fingerprints and the verifier. Since a large number of fingerprints is typically required to ensure high verification reliability, this training architecture introduces a substantial number of jointly trained parameters. Consequently, the risk of overfitting to limited training models in both sets is further amplified, potentially undermining MetaV's generalization performance.

In this paper, we propose LiteGuard, an efficient task-agnostic model fingerprinting method that enhances generalization while maintaining low computational cost. To achieve this, LiteGuard introduces the following two key innovations: (i) Checkpoint-based model set augmentation: Lite-Guard leverages checkpoints—intermediate model snapshots saved during the training (Eisenman et al., 2022; Wang et al., 2021a) of each pirated and independently-trained model—to augment the pirated and independently-trained model sets. These checkpoints reflect diverse decision behaviors, enhancing model diversity at no extra computational cost; (ii) Local verifier architecture: Instead of using a global verifier that is jointly trained with all the fingerprints, LiteGuard pairs each fingerprint with a lightweight local verifier. While each pair is jointly trained, different pairs are optimized independently, substantially reducing the number of jointly trained parameters. The two novel designs aim to address the overfitting issue caused by reducing the number of training models, thereby enhancing the computational efficiency of the task-agnostic fingerprinting paradigm without compromising generalization. We evaluate LiteGuard on five representative tasks involving multiple types of DNN models. Experimental results demonstrate that LiteGuard consistently surpasses both task-agnostic and task-specific fingerprinting baselines, achieving higher generalization capability and computational efficiency.

## 2 PROBLEM FORMULATION

We consider a typical model fingerprinting scenario involving two entities: a model owner and an adversary, illustrated in Figure 1. Specifically, the model owner trains and deploys a proprietary DNN model $M_P$. To safeguard its ownership, the model owner generates a set $\mathcal{F}$ of fingerprints by crafting input samples that elicit distinctive responses from $M_P$. Meanwhile, an adversary gains unauthorized access to $M_P$, e.g., via model theft, and redistributes this pirated copy as a black-box service that clients can access through an API. In particular, the adversary may apply performance-preserving ownership obfuscation techniques to the stolen model before redistribution, in order to evade ownership verification (Pan et al., 2022; Zhao et al., 2024; Xu et al., 2024).

Upon identifying a suspect model $M_S$, the model owner performs ownership verification to determine whether it is a piracy version of $M_P$. Concretely, the model owner queries $M_S$ using the fingerprint set $\mathcal{F}$ and analyzes its outputs to determine whether $M_S$ is a piracy version of $M_P$ or an independently-trained model.

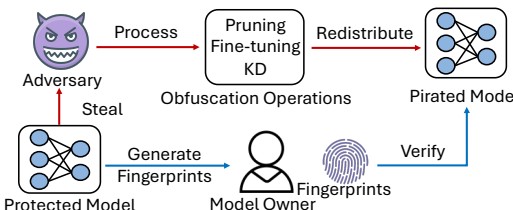

Figure 1: A typical model fingerprinting scenario.

## 3 METHODOLOGY

### 3.1 OVERVIEW

We present LiteGuard, a task-agnostic model fingerprinting framework with enhanced generalization capability under the limited computational cost. Specifically, LiteGuard jointly learns a set of fingerprint-verifier pairs through an end-to-end training process. As illustrated in Figure 2, LiteGuard consists of the following three components.

- **Model Set Construction.** LiteGuard constructs two model sets: an independence set and a piracy set. The independence set consists of models independently trained from scratch, along with the checkpoints captured during their training. The piracy set includes the protected model, its checkpoints, and optionally its variants with their checkpoints. By leveraging these checkpoints, LiteGuard significantly increases model diversity without requiring additional training efforts, thereby enhancing generalization capability at no extra computational cost.

- **Fingerprint-Verifier Pair Joint Training.** Each fingerprint and its paired local verifier are jointly trained using the constructed model sets. Unlike a global verifier that processes the joint responses from a model to all fingerprints, each local verifier operates solely on the models response to a single fingerprint, significantly reducing the number of jointly trained parameters and thus mitigating overfitting.

- **Ownership Verification.** During model ownership verification, each fingerprint is passed through the suspect model to obtain an output, which is evaluated by its corresponding local verifier to produce a confidence score. The final verification decision is made by averaging the confidence scores across all verifiers.

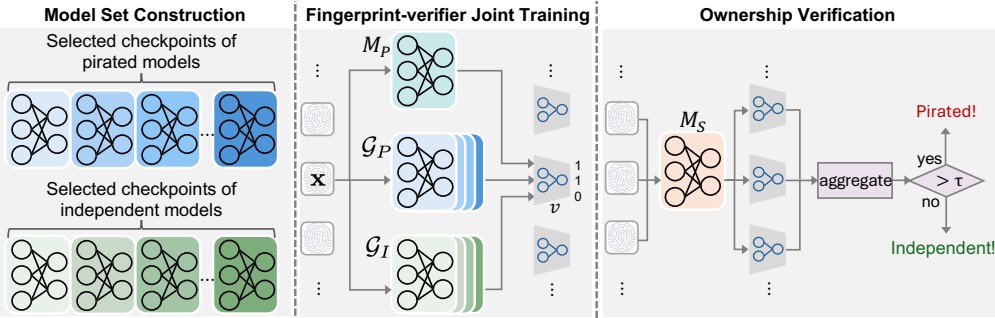

Figure 2: The overview of LiteGuard.

## 3.2 MODEL SET CONSTRUCTION

Prior to fingerprint generation, we construct two distinct model sets: a *piracy set* $\mathcal{G}_P$ and an *independence set* $\mathcal{G}_I$. The piracy set contains a protected model and may also include its variants produced by applying ownership obfuscation operations. In contrast, the independence set comprises models that are independently trained from scratch on the same or similar datasets, but with different seeds and/or architectures, thereby ensuring independence from the protected model. To introduce additional diversity for both sets, we capture model checkpoints—intermediate snapshots saved during training models for both sets—and incorporate them into their respective sets.

**The Checkpoint Selection Strategy.** A typical training process yields a large number of model checkpoints, but using all of them for fingerprint generation is resource-consuming. and often redundant. Thus, we propose a principled checkpoint selection strategy that maintains model diversity while reducing resource compsumption. Specifically, starting from an epoch $e_s$, we uniformly sample checkpoints throughout training, selecting one every $l$ epochs until the end of training. Choosing $e_s$ is crucial: checkpoints from the early stage often produce near-random predictions and lack meaningful decision behavior, while those near the final epoch tend to resemble the converged model and thus contribute little additional diversity. This approach reduces redundancy among adjacent checkpoints while ensuring sufficient diversity.

## 3.3 JOINT TRAINING OF FINGERPRINT-VERIFIER PAIRS

At this stage, we aim to construct a set of $N$ fingerprint-verifier pairs, each consisting of a fingerprint and a lightweight verifier. These pairs are trained using the piracy set $\mathcal{G}_P$ and the independence set $\mathcal{G}_I$, with the objective of maximizing their ability to distinguish pirated models from independently-trained ones. To mitigate overfitting, we adopt batch-wise training: At each iteration, a fingerprint-verifier pair is optimized using two mini-batches of $K$ models, denoted as $\mathcal{B}_P$ and $\mathcal{B}_I$, which are randomly sampled from $\mathcal{G}_P$ and $\mathcal{G}_I$, respectively.

Let $\mathbf{x} \in \mathcal{X}$ denote a trainable fingerprint, where $\mathcal{X}$ is the input space of the models. For each model $g \in \mathcal{B}_P \cup \mathcal{B}_I$, the model output is computed as $\mathbf{y} = g(\mathbf{x})$. This output is then evaluated by a corresponding verifier $v : \mathcal{Y} \to (0, 1)$, where $\mathcal{Y}$ is the output space of the models. The verifier $v$ is designed as a lightweight model comprising a linear layer followed by a sigmoid activation. Specifically, given a model output $\mathbf{y} \in \mathcal{Y}$, the verifier computes $v(\mathbf{y}) = \sigma(\langle \mathbf{w}, \mathbf{y} \rangle_F)$, where $\mathbf{w}$ is a trainable weight tensor of the same dimensionality as $\mathbf{y}$, $\langle \cdot, \cdot \rangle_F$ represents the Frobenius inner product, and $\sigma(\cdot)$ is the sigmoid function. A binary label $s$ is assigned to each model, where $s = 1$ indicates pirated models and $s = 0$ indicates independently-trained models.

The fingerprint $\mathbf{x}$ and verifier $v$ are jointly trained with the objective of minimizing the average binary cross-entropy loss over the pair of sampled model batches. Specifically, the objective function is defined as follows:

$$\mathcal{L}(\mathbf{x}, v) = \alpha_{prot} \cdot \mathcal{L}_{\text{bce}}(v(M_P(\mathbf{x})), 1) + \frac{\alpha_P}{K} \sum_{g_P \in \mathcal{B}_P} \mathcal{L}_{\text{bce}}(v(g_P(\mathbf{x})), 1) + \frac{\alpha_I}{K} \sum_{g_I \in \mathcal{B}_I} \mathcal{L}_{\text{bce}}(v(g_I(\mathbf{x})), 0),$$
(1)

where $\mathcal{L}_{\text{bce}}(p, s) = -s \log p - (1 - s) \log(1 - p)$ is the binary cross-entropy loss. The weights are chosen to satisfy $\alpha_{prot} + \alpha_P \approx \alpha_I$, ensuring a balanced contribution from positive and negative samples to prevent biased optimization. The parameters of both $\mathbf{x}$ and $v$ are updated using the Adam optimizer (Kingma & Ba, 2017) based on the gradients of the objective.

Upon the completion of training, we obtain the fingerprint set $\mathcal{F} : \{(\mathbf{x}_m^*, v_m^*)\}_{m=1}^N$, where each $(\mathbf{x}_m^*, v_m^*)$ represents a trained fingerprint-verifier pair.

## 3.4 OWNERSHIP VERIFICATION

At the ownership verification stage, the goal is to determine whether a suspect model is a pirated copy of the protected model $M_P$ or an independently-trained model. To this end, we leverage the learned fingerprint-verifier set $\mathcal{F} = \{(\mathbf{x}_m^*, v_m^*)\}_{m=1}^N$ to perform ownership verification.

Given a suspect model $M_S$, each fingerprint $\mathbf{x}_m^*$ is first fed into the model to produce an output $\mathbf{y}_m = M_S(\mathbf{x}_m^*)$, which is then evaluated by the corresponding verifier $v_m^*$, producing a confidence

score $v_m^*(\mathbf{y}_m) \in (0, 1)$ that indicates the likelihood of $M_S$ being a piracy version of the protected model $M_P$. Subsequently, the $N$ confidence scores corresponding to the $N$ fingerprint-verifier pairs are aggregated by taking their average, i.e., $s_{\text{avg}} = \frac{1}{N} \sum_{m=1}^{N} v_m^* (M_S(\mathbf{x}_m^*))$. After obtaining the averaged confidence score $s_{\text{avg}}$, a verification decision is made by comparing $s_{\text{avg}}$ with a threshold $\tau \in (0, 1)$. If $s_{\text{avg}} > \tau$, $M_S$ is determined as a pirated copy of $M_P$, otherwise $M_S$ is classified as an independently-trained model. Formally, the decision rule is defined by $\text{Decision}(M_S) = \mathbf{1}\,[s_{\text{avg}} > \tau]$, where $\mathbf{1}[\cdot]$ is the indicator function, which returns 1 if the condition holds, and 0 otherwise.

## 4 PARAMETER COMPLEXITY ANALYSIS

We analyze the parameter complexity of LiteGuard and MetaV. Here, parameter complexity is defined as the total number of learnable parameters entangled in the joint training architecture of each method. To facilitate this analysis, we introduce the following notation: Let $F$ denote the number of parameters per fingerprint, $O$ the model output dimension, and $N$ the total number of fingerprints.

**MetaV: Entangled Architecture.** MetaV adopts an entangled architecture in which all fingerprints are trained jointly with a global verifier. The global verifier is implemented as an $L$-layer multilayer perception (MLP) with $H$ neurons in each hidden layer. It takes as input the concatenated model outputs corresponding to all fingerprints, resulting in an input dimension of $N \cdot O$, and maps them to a scalar score. Hence, the number of parameters contained by the global verifier is $P_{\text{verifier}}^{\text{MetaV}} = (N \cdot O) \cdot H + (L - 2) \cdot H^2 + H$. Together with $N$ fingerprints, the number of entangled parameters that are jointly trained is $P_{\text{total}}^{\text{MetaV}} = N \cdot F + P_{\text{verifier}}^{\text{MetaV}}$, which leads to an overall complexity of $P_{\text{MetaV}} = \mathcal{O}\big(N(F + O \cdot H) + H^2 \cdot L\big)$.

**LiteGuard: Decoupled Architecture.** LiteGuard employs a modular design, where each fingerprint is paired with a lightweight, independent local verifier. Each fingerprint-verifier pair consists of $F$ parameters from the fingerprint and $O$ from the linear layer of the local verifier that maps the model's $O$-dimensional output to a scalar score. Thus, the number of jointly trained parameters is $P_{\text{pair}}^{\text{LiteGuard}} = F + O$. Since all pairs are optimized independently, the parameter complexity of this design remains constant, i.e., $P_{\text{LiteGuard}} = \mathcal{O}(F + O)$.

**Implications.** MetaV's parameter complexity scales with the number of fingerprints $N$, consequently increasing the risk of overfitting, particularly when $N$ is large. In contrast, LiteGuards decoupled architecture maintains constant parameter complexity independent of $N$, thereby mitigating the overfitting risk and improving generalization capability.

## 5 EXPERIMENTS

We perform systematic experiments to thoroughly evaluate the effectiveness of LiteGuard with the following experimental settings.

**Tasks.** We consider five typical tasks, each corresponding to a specific type of neural network architecture and a dataset: (1) image classification using convolutional neural networks (CNNs) on the CIFAR-100 dataset (Krizhevsky et al., 2014), (2) protein property regression using MLPs on the CASP dataset (Rana, 2013), (3) tabular data generation with autoencoders (AEs) on the California Housing (CH) dataset (Pace & Barry, 1997), (4) molecular property prediction via GNNs on the QM9 dataset (Ruddigkeit et al., 2012; Ramakrishnan et al., 2014), and (5) time-series sequence data generation with recurrent neural networks (RNNs) on the Weather dataset (NCEI, 2023).

**Baselines.** We compare LiteGuard with several representative model fingerprinting methods. Specifically, since MetaV is a task-agnostic approach, we compare LiteGuard with MetaV across all five tasks. Besides, for the molecular property prediction task, we also include GNNFingers (You et al., 2024), a method specifically tailored for GNNs. For the classification task, besides MetaV, we additionally compare LiteGuard with several representative existing methods, including IPGuard (Cao et al., 2021), which utilizes near-boundary samples that capture unique model behaviors as fingerprints, UAP (Peng et al., 2022), which is based on universal adversarial perturbations, and ADV-TRA (Xu et al., 2024), which employs adversarial trajectories that encode richer decision boundary information to achieve enhanced verification reliability.

Table 2: AUCs achieved by different approaches across five tasks (different model types and datasets).

| Method | CNN/CIFAR-100 | MLP/CASP | AE/CH | GNN/QM9 | RNN/Weather |
|---|---|---|---|---|---|
| UAP | $0.752 \pm 0.045$ | – | – | – | – |
| IPGuard | $0.708 \pm 0.055$ | – | – | – | – |
| ADVTRA | $0.845 \pm 0.010$ | – | – | – | – |
| GNNFingers | – | – | – | $0.608 \pm 0.165$ | – |
| MetaV | $0.676 \pm 0.019$ | $0.824 \pm 0.008$ | $0.783 \pm 0.029$ | $0.598 \pm 0.108$ | $0.854 \pm 0.022$ |
| **LiteGuard** | **$0.936 \pm 0.007$** | **$0.902 \pm 0.007$** | **$0.977 \pm 0.001$** | **$0.803 \pm 0.003$** | **$0.971 \pm 0.004$** |

**Ownership obfuscation techniques.** We consider six representative ownership obfuscation techniques: pruning, fine-tuning, knowledge distillation (KD), noise injection followed by fine-tuning (N-finetune), pruning followed by fine-tuning (P-finetune), and adversarial fine-tuning (A-finetune).

**Model set construction.** For each task, we designate a protected model, and construct four model sets to facilitate the training and testing of our method—each phase demanding both a piracy set and an independence set. Specifically, unless otherwise specified in experimental settings, we adopt an extreme setting for the training phase, wherein each model set contains only a single trained model: the piracy set consists of the protected model, while the independence set consists of one independently-trained model. This is the highest-efficiency scenario with minimal training overhead, enabling a direct evaluation of LiteGuards capability to mitigate overfitting and improve generalization. Furthermore, we augment both sets by incorporating checkpoints saved during the training process of the protected and independently-trained models. Table 1 summarizes the checkpoint selection configurations for each model type. For the testing phase, we construct large and diverse model sets that are entirely disjoint from those used in training: the piracy set comprises 91 models, including the protected model and 15 variants for each of six ownership obfuscation techniques, each generated using different random seeds, while the independence set consists of 100 models trained from scratch using varying architectures and random seeds. The testing model sets for experimental evaluation are completely disjoint from the model sets used for fingerprint generation, ensuring that the performance is assessed solely on previously unseen models.

**Evaluation Metric.** We employ the Area Under the Receiver Operating Characteristic Curve (AUC) as our primary evaluation metric when assessing the effectiveness of LiteGuard. More specifically, AUC measures the likelihood that a randomly-chosen pirated model is assigned a higher confidence score than a randomly-chosen independently-trained model, thereby quantifying the capability of a fingerprinting method to

Table 1: Configurations of checkpoints selection for each model type: start epoch $e_s$, selection interval $l$, and total training epochs $E$.

| | CNN | MLP | AE | GNN | RNN |
|---|---|---|---|---|---|
| $l$ | 20 | 6 | 6 | 12 | 6 |
| $e_s/E$ | 400/600 | 60/120 | 60/120 | 180/300 | 60/120 |

discriminate pirated models and independently-trained models. A higher AUC value generally indicates stronger verification performance, and an AUC of 1.0 represents perfect discrimination between pirated and independently-trained models. In the experiments, we report the mean and standard deviation of AUCs over five independent runs.

**Other Implementation Details.** To evaluate LiteGuard, we set the loss weights $\alpha_{prot}$, $\alpha_P$, and $\alpha_I$ to be 0.5, 1.0, and 1.5, respectively, and use a batch size $K = 1$. We set the number of fingerprints to $N = 100$ across all methods, consistent with the setting in MetaV (Pan et al., 2022). We train LiteGuard under the same training protocol used in MetaV, including using the Adam optimizer (Kingma & Ba, 2017) with a learning rate of 0.001 and 1000 training iterations.

## 5.1 EFFECTIVENESS OF OUR PROPOSED METHOD

**The Generalization Capability of LiteGuard.** We evaluate LiteGuards generalization capability by assessing its effectiveness in distinguishing unseen pirated models from unseen independently-trained models in the testing phase, under a highly constrained training setting where only a single model is used in each model set during fingerprint training. Table 2 presents the AUCs achieved by LiteGuard and various baselines across five representative tasks. Specifically, LiteGuard achieves the state-of-the-art performance on all five tasks, outperforming both task-specific baselines (i.e., UAP, IPGuard,

ADVTRA, and GNNFingers) and the task-agnostic baseline (i.e., MetaV). Particularly, LiteGuard, as a task-agnostic approach, surprisingly outperforms task-specific methods that are specifically tailored for their target tasks, underscoring its consistent effectiveness across different domains. Compared to MetaV, LiteGuard yields substantial improvement in AUC: 34.5% on the image classification task, 9.5% on the protein property regression task, 24.9% on the tabular data generation task, 34.3% on the molecular prediction task, and 13.7% on the time-series sequence generation task. These results demonstrate LiteGuard's significantly enhanced generalizability.

We further plot the Receiver Operating Characteristic (ROC) curves and the distribution of confidence scores to illustrate the discriminative power of LiteGuard. The ROC curve depicts the relationship between true positive rate (TPR) and false positive rate (FPR) across different thresholds, offering a holistic view of a methods ability to distinguish pirated models from independent ones. A curve closer to the top-left corner indicates stronger discriminative power. As shown in Figure 3(a), LiteGuard achieves the most favorable ROC profile on the image classification task, closely approaching the ideal top-left point and outperforming all baselines. In contrast, MetaV's ROC curve lies much closer to the diagonal, reflecting its limited discriminative power under the constrained training setting. On the other hand, we also present the distribution of confidence scores for pirated and independently-trained models through boxplots in Figure 3(b). A larger separation between the score distributions of pirated and independently-trained models implies higher discriminative power of a method. LiteGuard yields two clearly separated distributions with minimal overlap, indicating a strong ability to distinguish pirated models from independently-trained ones. By comparison, other methods—especially MetaV—exhibit significant overlap, revealing their limited ability to reliably verify model ownership.

**The Computational Efficiency of LiteGuard.**

Figure 4 demonstrates LiteGuard's computational efficiency by measuring the mean AUC (depicted as a solid line) and the standard deviation (shown as a shaded area) across varying numbers of trained models in each model set used for training fingerprint-verifier pairs. We evaluate efficiency with respect to the number of trained models because such training operations dominate the computation cost for both MetaV and LiteGuard. Our comparison focuses on two tasks: the molecular property prediction task (GNN/QM9) and the protein property regression task (MLP/CASP). For LiteGuard, we vary the number of trained models in both sets from 1 to 5 and augment each set with checkpoints to ensure a fixed size of 10 models, which incur negligible additional cost. For MetaV, we consider three configurations with n = 1, n = 10, and n = 30 trained models per set.

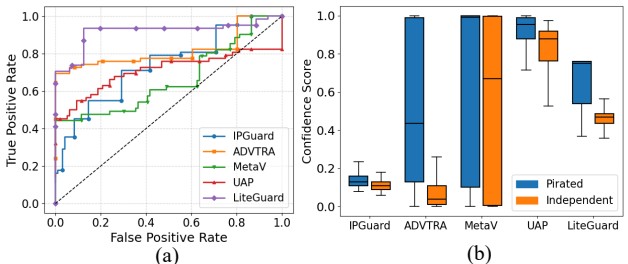

Figure 3: (a) The ROC curve and (b) the confidence score distribution.

As shown in Figure 4(a), on the molecular property prediction task, LiteGuard achieves an AUC of 0.858 using 2 trained models per set, outperforming MetaV's AUC of 0.849 obtained using 10 trained models per set, this corresponds to around 5×fewer model training cost. With 5 trained models, LiteGuard's AUC further improves to 0.944, nearly matching MetaV's performance at n = 30–a 6×model training cost reduction. A similar trend is observed in

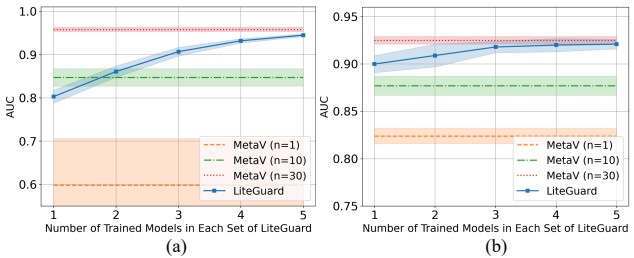

Figure 4: The AUCs achieved by MetaV and LiteGuard on (a) the molecular property prediction task (GNN/QM9) and (b) the protein property regression task (MLP/CASP).

Figure 4(b) for the protein property regression task, where LiteGuard with even 1 trained model outperforms MetaV's performance at n = 10 (10×reduction), and with 5 models, matches MetaV's AUC at n = 30 (6×reduction). These results demonstrate that LiteGuard achieves superior owner-

Table 3: AUCs under various ownership obfuscation techniques across various tasks.

| Model/Dataset | Method | Pruning | Finetuning | KD | N-finetune | P-finetune | A-finetune |
|---|---|---|---|---|---|---|---|
| CNN/CIFAR100 | UAP | 0.731 ± 0.068 | 0.903 ± 0.003 | 0.703 ± 0.072 | 0.642 ± 0.054 | 0.752 ± 0.050 | 0.780 ± 0.066 |
| | IPGuard | 0.742 ± 0.078 | 0.990 ± 0.011 | 0.581 ± 0.030 | 0.720 ± 0.024 | 0.653 ± 0.017 | 0.561 ± 0.027 |
| | ADVTRA | 0.887 ± 0.011 | 0.993 ± 0.005 | **0.816 ± 0.008** | 0.726 ± 0.009 | 0.897 ± 0.015 | 0.869 ± 0.011 |
| | MetaV | 0.728 ± 0.030 | 0.776 ± 0.017 | 0.603 ± 0.022 | 0.707 ± 0.013 | 0.681 ± 0.012 | 0.511 ± 0.018 |
| | **LiteGuard** | **0.955 ± 0.007** | **0.996 ± 0.002** | 0.786 ± 0.009 | **0.986 ± 0.005** | **0.975 ± 0.010** | **0.980 ± 0.004** |
| MLP/CASP | MetaV | **0.997 ± 0.005** | 0.988 ± 0.010 | 0.520 ± 0.038 | 0.987 ± 0.015 | 0.697 ± 0.070 | 0.886 ± 0.056 |
| | **LiteGuard** | 0.981 ± 0.013 | **1.000 ± 0.000** | **0.534 ± 0.004** | **1.000 ± 0.000** | **0.985 ± 0.003** | **0.978 ± 0.002** |
| AE/CH | MetaV | 0.655 ± 0.015 | 0.826 ± 0.051 | 0.798 ± 0.022 | 0.812 ± 0.025 | 0.779 ± 0.022 | 0.801 ± 0.005 |
| | **LiteGuard** | **0.927 ± 0.005** | **0.992 ± 0.000** | **0.991 ± 0.000** | **0.991 ± 0.001** | **0.975 ± 0.001** | **0.991 ± 0.000** |
| GNN/QM9 | GNNFingers | 0.688 ± 0.111 | 0.637 ± 0.266 | 0.506 ± 0.105 | 0.589 ± 0.196 | 0.621 ± 0.125 | 0.623 ± 0.031 |
| | MetaV | 0.673 ± 0.047 | 0.601 ± 0.162 | 0.550 ± 0.020 | 0.580 ± 0.164 | 0.681 ± 0.098 | 0.601 ± 0.022 |
| | **LiteGuard** | **0.984 ± 0.001** | **0.834 ± 0.005** | **0.645 ± 0.005** | **0.791 ± 0.003** | **0.812 ± 0.004** | **0.770 ± 0.002** |
| RNN/Weather | MetaV | 0.870 ± 0.018 | 0.816 ± 0.030 | 0.922 ± 0.018 | 0.835 ± 0.016 | 0.835 ± 0.023 | 0.762 ± 0.021 |
| | **LiteGuard** | **0.988 ± 0.000** | **0.964 ± 0.007** | **0.991 ± 0.003** | **0.962 ± 0.005** | **0.957 ± 0.006** | **0.947 ± 0.002** |

ship verification performance while requiring much fewer trained models, thus yielding substantial computational savings.

**Robustness to Ownership Obfuscation Techniques.**

We evaluate the robustness of LiteGuard against a range of ownership obfuscation techniques (i.e., pruning, fine-tuning, KD, N-finetune, P-finetune, and A-finetune). The corresponding results are summarized in Table 3. It can be observed that LiteGuard consistently outperforms all baseline methods in most cases, demonstrating strong robustness against various ownership obfuscation techniques. Specifically, for the image classification task (CNN/CIFAR-100), LiteGuard achieves near-perfect verification performance under four of the six techniques, achieving AUCs of 0.996 (Finetuning), 0.986 (N-finetune), 0.975 (P-finetune), and 0.980 (A-finetune). It also maintains strong performance under pruning (0.955) and exhibits competitive robustness against the more challenging KD (0.786). Particularly, compared to MetaV, LiteGuard shows substantial improvements in AUC under all six obfuscation techniques—namely, 31.1% (Pruning), 28.4% (Finetuning), 30.3% (KD), 39.5% (N-finetune), 43.2% (P-finetune), and 91.8% (A-finetune). Beyond the image classification task, LiteGuard also demonstrates superior robustness across the other four tasks. For instance, on the tabular data generation task (AE/CH), LiteGuard achieves 41.5% higher AUC than MetaV under pruning, and on the molecular property prediction task (GNN/QM9), it outperforms MetaV by 38.8% under fine-tuning. An exception is observed in the protein property regression task (MLP/CASP) under pruning, where both LiteGuard and MetaV yield comparable AUCs. These results underscore LiteGuard's strong robustness against a broad range of ownership obfuscation techniques.

## 5.2 ABLATION STUDIES

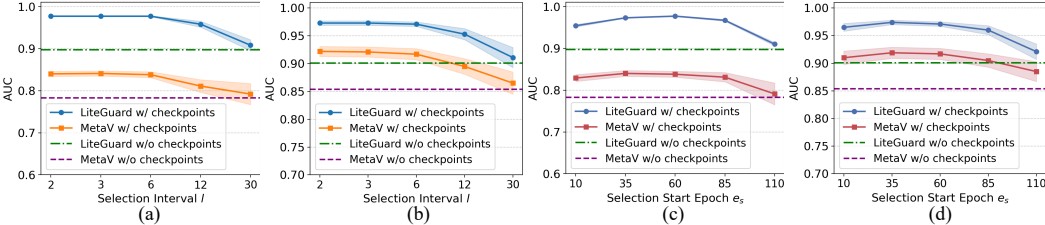

Figure 5: The AUCs under varying checkpoint selection intervals $l$ for (a) tabular data generation task (AE/CH) and (b) time-series sequence generation task (RNN/Weather). The AUCs under varying start epoch $e_s$ for (c) tabular data generation task (AE/CH) and (d) time-series sequence generation task (RNN/Weather).

**Impact of the Checkpoint-based Augmentation.** We evaluate the impact of checkpoint-based augmentation and its configuration parameters on discriminative capability. More specifically, we vary the selection interval $l$ and the starting epoch $e_s$. Besides, to assess whether MetaV can benefit from incorporating checkpoints, we also extend its implementation with checkpoint augmentation. The experimental results are shown in Figure 5, in which solid lines represent methods with checkpoint augmentation, while dashed lines represent the corresponding methods without it.

Figure 5(a), (b), (c), and (d) show the impact of varying selection interval $l$ and varying selection start epoch $e_s$ on the tabular data generation task and the time-series sequence generation task, respectively. We can see from the sub-figures that for both LiteGuard and MetaV, the solid lines are always above the corresponding dashed lines, demonstrating that introducing extra checkpoints to the piracy and independence sets increases diversity and thus enhances discriminative power. These results suggest that checkpoint-based augmentation is a generally effective strategy for enhancing generalization capability.

Figure 5(a) and (b) show the impact of varying selection interval $l$ with fixed start epoch $e_s$ on both tasks. A clear decline in AUC is observed when $l$ increases from 6 to 30, as a larger interval yields fewer available checkpoints and thereby limits the diversity of the model set. In contrast, when the interval decreases from 6 to 2, the performance quickly saturates since close checkpoints provide limited additional diversity.

Figure 5(c) and (d) show the impact of varying selection start epoch $e_s$ with fixed interval $l$ at 2 on both tasks. Adopting an early start epoch (e.g., $e_s = 10$) lowers AUCs because early-stage models behave almost randomly and lack meaningful decision behaviors. Using a late start epoch (e.g., $e_s = 110$) also exhibits a similar performance degradation, as such checkpoints are close to the converged model and contribute little diversity. The best results occur with moderate $e_s$ values (e.g., 3585), which avoid noisy early checkpoints while retaining sufficiently diverse intermediate ones. Particularly, the AUCs achieved when $e_s = 10$ are always higher than those achieved when $e_s = 110$, since an early start still includes mid- and late-stage checkpoints and thus preserves sufficient diversity, whereas a late start is restricted to overly similar checkpoints.

**Impact of the Verifier Design.** Figure 6 shows the effect of the verifier design on the discriminative power across two tasks. In the experiment, we consider four configurations: Lite-Guard, MetaV, LiteGuard-MV (Lite-Guard using MetaVs verifier), and MetaV-LV (MetaV using LiteGuards verifier). As shown in the figure, Lite-Guard achieves the highest AUC on both tasks, outperforming all other configurations. On the image classification task, LiteGuard attains an AUC of 0.936, substantially exceed-

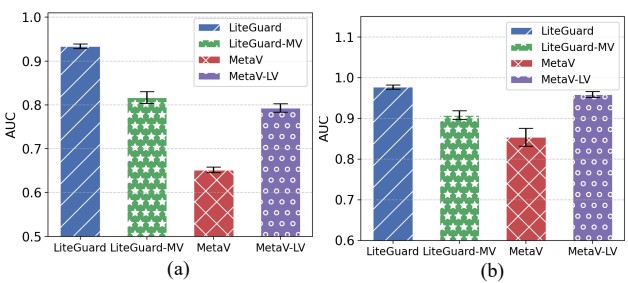

Figure 6: The AUCs under different verifier designs for (a) the image classification task (CNN/CIFAR-100) and (b) the time-series sequence data generation task (RNN/Weather).

ing LiteGuard-MV (0.809), MetaV (0.676), and MetaV-LV (0.793). A similar trend is observed in the time-series task, where LiteGuard achieves an AUC of 0.977, while LiteGuard-MV drops to 0.908. MetaV achieves an AUC of 0.854, which is substantially improved to 0.959 when equipped with LiteGuard's verifier. Therefore, these results clearly reveal the importance of LiteGuard's verifier design in enhancing discriminative capability.

**Impact of the Fingerprint Set Size.** Figure 7 demonstrates the impact of the fingerprint set size on the verification performance. As previously discussed, MetaV's performance is highly sensitive to the number of fingerprints. To assess whether Lite-Guard exhibits similar behavior, we conduct experiments on the tabular data generation task and the protein property regression task. The experimental results in Figure 7 reveal that LiteGuard achieves a monotonic in-

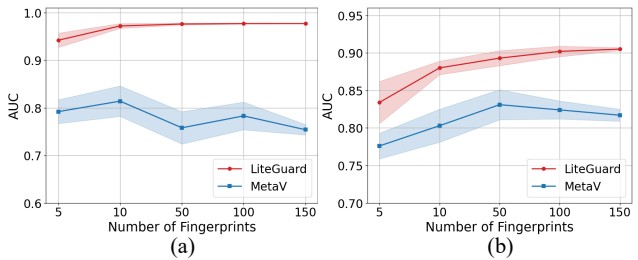

Figure 7: The AUC under varying numbers of fingerprints for (a) the tabular data generation task (AE/CH) and (b) the protein property regression task (MLP/CASP).

crease in AUC as the number of fingerprints grows, with consistently decreasing standard deviations, indicating stable performance improvements. This observation reveals that LiteGuard benefits from the increased number of fingerprints without suffering from overfitting. In contrast, MetaV shows a fluctuating and non-monotonic trend. While it initially gains from a moderate increase in the number

of fingerprints, its performance deteriorates beyond a certain point. This suggests that MetaV's entangled fingerprints-global verifier design introduces susceptibility to overfitting when the fingerprint set becomes large. Consequently, MetaV requires careful tuning of the fingerprint set size, which is non-trivial in practice.

## 6 DISCUSSION: A UNIFIED VERIFIER

MetaV's global verifier and LiteGuards local verifiers represent two extremes along a broader design spectrum for aggregating fingerprints for ownership decisions. A natural approach is to design a *unified verifier* paradigm: partition the fingerprint set into multiple groups, with each group sharing a lightweight verifier that jointly processes the concatenated outputs of the fingerprints within that group. This design retains the ability to model cross-fingerprint interactions through joint verification, while mitigating the full parameter coupling and overfitting risks that arise when the available model pool is limited. At the same time, it generalizes the fully decoupled local-verifier strategy by allowing controlled coupling within groups, offering a practical mechanism to balance generalization, robustness, and computational efficiency.

## 7 CONCLUSIONS

In this paper, we present LiteGuard, a task-agnostic model fingerprinting method that offers enhanced generalization capability under high-efficiency settings where the number of trained models in the piracy and independence sets is limited, addressing the generalization issue suffered by MetaV. By leveraging checkpoint-based model set augmentation and a modular local verifier architecture, LiteGuard significantly improves model diversity and verification robustness without incurring extra computational costs. Experimental results across various tasks demonstrate that LiteGuard achieves superior verification performance, outperforming MetaV in both generalization performance and computational efficiency. These results highlight LiteGuard's practical value as an efficient task-agnostic fingerprinting method.

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

Table 4: 95% confidence intervals (CI) for AUCs by different approaches across five tasks (different model types and datasets).

| Method | CNN/CIFAR-100 | MLP/CASP | AE/CH | GNN/QM9 | RNN/Weather |
|---|---|---|---|---|---|
| UAP | [0.696, 0.808] | – | – | – | – |
| IPGuard | [0.640, 0.776] | – | – | – | – |
| ADVTRA | [0.833, 0.857] | – | – | – | – |
| GNNFingers | – | – | – | [0.403, 0.813] | – |
| MetaV | [0.652, 0.699] | [0.814, 0.834] | [0.747, 0.819] | [0.464, 0.732] | [0.827, 0.881] |
| LiteGuard | [0.927, 0.945] | [0.893, 0.911] | [0.976, 0.978] | [0.799, 0.807] | [0.966, 0.976] |

Table 5: 95% confidence intervals (CI) for AUCs under various ownership obfuscation techniques across various tasks.

| Dataset | Method | Pruning | Finetuning | KD | N-Finetune | P-Finetune | A-Finetune |
|---|---|---|---|---|---|---|---|
| CNN/CIFAR-100 | UAP | [0.647, 0.815] | [0.899, 0.907] | [0.614, 0.792] | [0.575, 0.709] | [0.690, 0.814] | [0.698, 0.862] |
| | IPGuard | [0.645, 0.839] | [0.976, 1.004] | [0.544, 0.618] | [0.690, 0.750] | [0.632, 0.674] | [0.527, 0.595] |
| | ADVTRA | [0.873, 0.901] | [0.987, 0.999] | [0.806, 0.826] | [0.715, 0.737] | [0.878, 0.916] | [0.855, 0.883] |
| | MetaV | [0.691, 0.765] | [0.755, 0.797] | [0.576, 0.630] | [0.691, 0.723] | [0.666, 0.696] | [0.489, 0.533] |
| | **LiteGuard** | **[0.946, 0.964]** | **[0.994, 0.998]** | **[0.775, 0.797]** | **[0.980, 0.992]** | **[0.963, 0.987]** | **[0.975, 0.985]** |
| MLP/CASP | MetaV | [0.991, 1.003] | [0.976, 1.000] | [0.473, 0.567] | [0.968, 1.006] | [0.610, 0.784] | [0.816, 0.956] |
| | **LiteGuard** | **[0.965, 0.997]** | **[1.000, 1.000]** | **[0.529, 0.539]** | **[1.000, 1.000]** | **[0.981, 0.989]** | **[0.976, 0.980]** |
| AE/CH | MetaV | [0.636, 0.674] | [0.763, 0.889] | [0.771, 0.825] | [0.781, 0.843] | [0.752, 0.806] | [0.795, 0.807] |
| | **LiteGuard** | **[0.921, 0.933]** | **[0.992, 0.992]** | **[0.991, 0.991]** | **[0.990, 0.992]** | **[0.974, 0.976]** | **[0.991, 0.991]** |
| GNN/QM9 | GNNFingers | [0.550, 0.826] | [0.307, 0.967] | [0.376, 0.636] | [0.346, 0.832] | [0.466, 0.776] | [0.585, 0.661] |
| | MetaV | [0.615, 0.731] | [0.400, 0.802] | [0.525, 0.575] | [0.376, 0.784] | [0.559, 0.803] | [0.574, 0.628] |
| | **LiteGuard** | **[0.983, 0.985]** | **[0.828, 0.840]** | **[0.639, 0.651]** | **[0.787, 0.795]** | **[0.807, 0.817]** | **[0.768, 0.772]** |
| RNN/Weather | MetaV | [0.848, 0.892] | [0.779, 0.853] | [0.900, 0.944] | [0.815, 0.855] | [0.806, 0.864] | [0.736, 0.788] |
| | **LiteGuard** | **[0.988, 0.988]** | **[0.955, 0.973]** | **[0.987, 0.995]** | **[0.956, 0.968]** | **[0.950, 0.964]** | **[0.945, 0.949]** |

## APPENDIX

## A  THE CONFIDENCE INTERVALS FOR AUC RESULTS

Table 4 presents the 95% Confidence Intervals (CIs) for the AUCs of LiteGuard and various baselines across five representative tasks, and Table 5 summarizes the 95% CIs for the AUCs of LiteGuard and various baselines across a range of ownership obfuscation techniques. From both tables, we can observe that LiteGuard's confidence intervals do not overlap with those of MetaV and other baselines across all datasets and obfuscation settings. This observation highlights that LiteGuard's performance improvement is statistically significant.

## B  A UNIFIED VERIFIER DESIGN

We assess the unified verifier design examining how its performance changes as the number of fingerprints per group varies. Specifically, we divide the $N$ fingerprints into multiple groups of size $M$, assigning one verifier to each group. When $M = 1$, the design reduces to LiteGuards local-verifier architecture; when $M = N$, it becomes MetaVs global verifier. Each group-level verifier is trained jointly with the $M$ fingerprints in its group and takes as input the concatenated outputs of these fingerprints to produce the corresponding ownership decision.

We evaluate this unified framework on the CNN/CIFAR-100 task by fixing $N = 100$ and varying the group size $M \in \{1, 2, 5, 10, 50\}$. The experimental results are summarized in Table 6. Empirically, we observe that partially shared verifiers do not outperform fully independent local verifiers, evidenced by the highest AUC obtained at $M = 1$. As $M$ increases, i.e., more fingerprints sharing the same verifier, AUC gradually decreases due to the reintroduction of jointly optimized parameters, which increases overfitting risk when only a small number of training models is available.

Table 6: AUC under different group sizes.

| $M$ | AUC |
|---|---|
| 1 | $0.936 \pm 0.007$ |
| 2 | $0.932 \pm 0.007$ |
| 5 | $0.925 \pm 0.011$ |
| 10 | $0.906 \pm 0.014$ |
| 50 | $0.853 \pm 0.023$ |

## C  DISCUSSION ON BIAS AND VARIANCE

We discuss how checkpoint-based augmentation and local verifier design influence bias and variance. Checkpoint-based augmentation increases the diversity of model behaviors that each fingerprint-verifier pair encounters during training, which makes the learned decision rule less sensitive to any single model and therefore reduces variance. Modularity further lowers variance by reducing the number of jointly trained parameters, which is crucial when the number of available training models is limited. Bias is not the limiting factor in this setting, as the fingerprint-verifier pair already offers sufficient expressive power for the binary discrimination task. This is evidenced by the experimental results on GNN/QM9 summarized in Table 7, where deeper verifier architectures do not yield performance gains. Instead, they reduce AUC, indicating that the added capacity increases the number of jointly optimized parameters and ultimately harms generalization.

Table 7: Impact of verifier depth on the verification performance.

| Verifier Depth | AUC (QM9) |
|---|---|
| 1 layer | $0.803 \pm 0.003$ |
| 2 layers | $0.792 \pm 0.007$ |
| 3 layers | $0.766 \pm 0.021$ |
| 4 layers | $0.725 \pm 0.078$ |

## D  THE EFFICIENCY ANALYSIS

The execution time of task-agnostic fingerprinting comes from two sources: the query-time overhead and the training time. For the query-time overhead, same as MetaV, LiteGuard also requires exactly one forward pass per fingerprint, and its verification consists of a simple aggregation of local-verifier outputs, leading to negligible computational cost. Instead, the training time is the dominant execution cost in task-agnostic fingerprinting, as pirated and independently-trained models need to be trained for constructing the two model sets.

Therefore, LiteGuard's computational efficiency stems from its ability to dramatically reduce the number of trained models required. Figure 4 highlights these savings. On the QM9 task, LiteGuard achieves an AUC of 0.858 with only 2 trained models per set, surpassing MetaV's AUC of 0.849 obtained with 10 trained models—a 5×reduction in training cost. With 5 trained models, LiteGuard reaches an AUC of 0.944, closely matching MetaV's performance with 30 models, yielding a 6×reduction. A similar pattern appears on the CASP regression task: LiteGuard with just 1 trained model exceeds MetaV's performance at 10 models (a 10×reduction), and with 5 models matches MetaV at 30 models (a 6×reduction). These results demonstrate that LiteGuard achieves MetaV-level performance while requiring far fewer trained modelsthe dominant source of computational cost.

## E  KEY DIFFERENCES BETWEEN LITEGUARD AND PRIOR WORKS

Table 8 summarizes the comparison of fingerprint methods along three key axes, including task-agnostic, model-set training cost, and verifier decoupling.

Table 8: Comparison of fingerprinting methods along three key axes.

| Method | Task-agnostic? | Model-set training cost | Scalability |
|---|---|---|---|
| IPGuard | No | – | High |
| UAP | No | Low–Moderate | High |
| ADVTRA | No | – | High |
| GNNFingers | No | High | Low |
| MetaV | Yes | High | Low |
| LiteGuard (ours) | Yes | Low | High |

The table shows that only MetaV and LiteGuard are task-agnostic, i.e., applicable for different tasks. Compared with MetaV, LiteGuard requires much lower model-set training cost and stronger scalability. In particular, the decoupled verifier design makes LiteGuard highly scalable—new fingerprint-verifier pairs can be added without retraining existing ones, and verification can flexibly adapt to different query budgets by using a corresponding number of fingerprint-verifier pairs. In contrast, MetaV requires retraining all fingerprints and the verifier together whenever the fingerprint set changes.

# F    IMPLEMENTATION DETAILS

## F.1    MODEL ARCHITECTURES AND TRAINING SETTINGS

We summarize the architectures and training configurations of the protected model, the pirated models, and the independently-trained models used for each task in the following.

### F.1.1    PROTECTED MODEL

**CNN (CIFAR-100).** The protected model is a ResNet-18 trained from scratch using SGD with momentum 0.9, weight decay 5e-4, learning rate 0.1, cosine annealing learning rate scheduling, and Xavier initialization. Training is conducted for 600 epochs with a batch size of 128.

**MLP (CASP).** The protected model is a Residual Multilayer Perceptron (ResMLP) trained from scratch using Adam Optimizer, with a learning rate of 0.001, and Kaiming initialization. Training is conducted for 120 epochs with a batch size of 128.

**GNN (QM9).** The protected model is a Graph Attention Network (GAT) trained using the Adam optimizer with a learning rate of 0.001 and Xavier initialization. Training is conducted for 300 epochs with a batch size of 128.

**AE (CH).** The protected model is a Variational Autoencoder (VAE) trained using the Adam optimizer with a learning rate of 0.01. In line with the default PyTorch implementation, the initialization scheme sets the weights of all linear layers to samples from a normal distribution $\mathcal{N}(0, 0.02^2)$ and their biases to zero. Training is conducted for 160 epochs with a batch size of 128.

**RNN (Weather).** The protected model is a Vanilla Recurrent Neural Network (RNN) trained using the Adam optimizer with a learning rate of 0.001 and Xavier initialization. Training is conducted for 120 epochs with a batch size of 128.

### F.1.2    INDEPENDENCE SET FOR FINGERPRINT TRAINING

For each task, the independence set used for fingerprint generation consists of a single independently-trained model together with its checkpoints. The architecture of the independently-trained model is: **CNN (CIFAR-100)**: DenseNet-169; **MLP (CASP)**: TabNet; **GNN (QM9)**: Graph Isomorphism Network (GIN); **AE (CH)**: Variational Autoencoder (VAE); **RNN (Weather)**: Gated Recurrent Unit (GRU).

### F.1.3 PIRACY SET FOR FINGERPRINT TRAINING

For each task, as mentioned in Section 5, the piracy set used for fingerprint generation only consists of the protected model and its checkpoints.

### F.1.4 INDEPENDENCE SET FOR PERFORMANCE EVALUATION

Across all tasks, models in the independence set are trained from scratch without access to the protected model; they share the tasks default training protocol for fairness, while diversity arises from heterogeneous architectures and distinct random seeds. The architectures included in the independence set for testing are specified as follows:

**CNN (CIFAR-100).** ResNet-18, ResNet-50, ResNet-101, MobileNetV2, MobileNetV3-Large, EfficientNet-B2, EfficientNet-B4, DenseNet-121, and DenseNet-169.

**AE (CH).** Autoencoder (AE), Variational Autoencoder (VAE), Wasserstein Autoencoder (WAE), Beta-Variational Autoencoder (BetaVAE), Sparse Autoencoder (SparseAE), Denoising Autoencoder (DenoisingAE), Adversarial Autoencoder (AdversarialAE).

**MLP (CASP).** Wide & Deep Model (Wide&Deep), Residual Multi-Layer Perceptron (ResMLP), Feature Tokenizer Transformer (FT-Transformer), Spiking Neural Network (SNN), Neural Ordinary Differential Equation (NODE), TabNet, and TabTransformer.

**GNN (QM9).** Graph Convolutional Network (GCN), Graph Isomorphism Network (GIN), Graph Sample and Aggregate (GraphSAGE), Graph Attention Network (GAT), Graph Attention Network v2 (GATv2), Simple Graph Convolution (SGC), and Approximate Personalized Propagation of Neural Predictions (APPNP).

**RNN (Weather).** Vanilla Recurrent Neural Network (VanillaRNN), Long Short-Term Memory (LSTM), Gated Recurrent Unit (GRU), Residual Recurrent Neural Network (Residual VanillaRNN), Residual Long Short-Term Memory (Residual LSTM), and Residual Gated Recurrent Unit (Residual GRU).

### F.1.5 PIRACY SET FOR PERFORMANCE EVALUATION

We generate pirated models for all tasks by applying six types of performance-preserving ownership obfuscation techniques to the protected model, with the following configurations:

- **Fine-tuning**: updating the parameters of the already trained protected model for a fixed number of epochs.

- **Adversarial fine-tuning (A-Finetune)**: fine-tuning the protected model using a mixture of clean and adversarially perturbed samples. For regression tasks with continuous outputs, adversarial examples are generated by maximizing the prediction loss (e.g., MSE) w.r.t. input perturbations using standard methods such as FGSM or PGD.

- **Pruning**: applying unstructured global pruning at sparsity levels ranging from 10% to 90% (step size 10%).

- **Prune + Fine-tuning (P-Finetune)**: pruning the model at sparsity levels of 30%, 60%, and 90%, followed by fine-tuning to recover performance.

- **Noise injection + Fine-tuning (N-Finetune)**: adding scaled Gaussian noise to the parameter tensor and then fine-tuning the noisy model.

- **Knowledge Distillation (KD)**: training a student model to mimic the protected model behavior by minimizing the KL divergence (or temperature-scaled soft targets) between the teacher and student outputs.

### F.2 DATASET PROCESSING

**Data Normalization.** Each dataset is preprocessed using its commonly adopted normalization parameters prior to model training, ensuring that inputs are scaled consistently with standard practice in the respective domains.

## G PRACTICAL DEPLOYMENT CONSIDERATIONS

In real-world deployments, a key consideration is the risk of false positives—namely, incorrectly flagging an independently trained model as pirated due to incidental behavioral similarity under a finite set of verification queries. This issue is practically important because a false-positive allegation may lead to erroneous IP claims, unnecessary legal or compliance procedures, and substantial reputational or operational costs for the parties involved. Importantly, this risk is inherent to ownership verification in general and is not specific to our method. Within our framework, false positives can be mitigated by adopting a more conservative verification threshold, which reduces the likelihood of accidental matches at the cost of decreased sensitivity.

Furthermore, fingerprinting-based ownership verification should be viewed as an evidence-generating mechanism rather than an definitive decision oracle. It functions as an initial screening tool that produces preliminary evidence. A positive screening outcome can then prompt more rigorous follow-up analysis, such as expanded testing with additional query sets, cross-validation with complementary evidence sources, or escalation to a formal third-party adjudication process when necessary.

## H RELATED WORK

Most existing studies exploit the characteristics of model decision boundaries to generate effective fingerprints for DNN models used for classification tasks. A common approach is to craft fingerprints that induce different labels for the protected model from other independent models. For instance, some researchers leverage adversarial examples as fingerprints to trigger distinguishable outputs for the protected models (Zhao et al., 2020; Yin et al., 2022; Peng et al., 2022; Yang & Lai, 2023). Cao et al. (2021) constructs near-boundary samples as fingerprints. Xu et al. (2024) forms adversarial trajectories to enhance verification reliability. Godinot et al. (2025) proposed to leverage the misclassified samples as fingerprints that demonstrate stronger discriminative power than normal adversarial examples. Tang et al. (2025) proposes a frequency-domain CNN fingerprinting scheme that embeds ownership information into low-frequency components of convolutional filters. Zhang et al. (2025) identifies the most informative triggers to achieve accurate model attribution under limited query budgets. However, the reliance on the presence of decision boundaries makes these methods applicable only to classification tasks.

Recent studies have begun to extend fingerprinting techniques to a broader set of tasks. For example, some methods (You et al., 2024; Waheed et al., 2024) generate fingerprints based on node features and graph topology to protect the ownership of graph neural network (GNN) models. Other studies (Yu et al., 2019; Huang et al., 2023) have explored training a classifier to extract unique identifiers from images produced by Generative Adversarial Network (GAN) models and used the identifiers as fingerprints for ownership verification. Fei et al. (2025) introduces a fingerprinting mechanism for latent diffusion models by encoding user-specific identifiers directly into model weights to enable model attribution.

While these task-specific fingerprinting techniques achieve strong performance within their intended domains, their dependence on task-tailored properties fundamentally limits their use across different tasks. This motivates the development of task-agnostic fingerprinting, which seeks a unified solution applicable to a broad range of models. To date, two approaches fall into this category: TAFA (Pan et al., 2021) and MetaV (Yang et al., 2022). TAFA imposes assumptions—such as requiring ReLU activations and continuous-valued outputs—that restrict its generality and prevent it from serving as a fully task-agnostic solution. MetaV employs a more universal strategy that co-trains a global verifier alongside a collection of fingerprints using both pirated and independently-trained models. However, MetaV's ability to generalize relies heavily on constructing large, diverse model sets, which incur substantial computational cost. While reducing the size of these model sets lowers the training cost, it also leads to significant drops in generalization performance, making it difficult to preserve reliable verification accuracy.

