# OpenReview forum: "LiteGuard: Efficient Task-Agnostic Model Fingerprinting with Enhanced Generalization"
_ICLR.cc/2026/Conference — ICLR 2026 Poster_

### Official Review · Reviewer_GCH5 · 2025-10-30

**Soundness:** 2
**Presentation:** 3
**Contribution:** 2
**Rating:** 4
**Confidence:** 4

**Summary:**

Liteguard targets the fingerprinting of DNNs (i.e., not LLMs) and places a focus on lightweight execution.

The idea of aggregating fingerprints and paired verifiers (the second component of the approach) is interesting, as verifiers are generally a single large model trained on all fingerprints.

**Strengths:**

* interesting verifier aggregating individual classifiers

**Weaknesses:**

* no guarantees whatsoever
* no proper related work
* missing discussions (open garden models, execution timings)

Liteguard is not backed by any formal guarantee, the whole performance assessment is experiment-based. And as the experimental section relies on modest scenarios in practice (cifar 100...), it questions the performance to be expected in modern applications. In particular, it is clear that performance depends on the models present and compared in both sets; experiments only go up to 5 or 30 models and variants per set, which is a tiny number today (see e.g., "Fingerprinting Classiﬁers with Benign Inputs",2023 where 1046 variants are discriminated, for larger DNNs), facing the proliferation of architectures, proposals, and potential variants. This precludes a clear forecast of practical performances and generalization to other included models.

Lacks consideration of unseen models: this is not very clear in the paper, but it seems that all experiments consider a "closed garden" setup, where only models at hand are considered (ie models that have been part of the training for discrimination). In practice, and as in multiple papers in the SoA, the question is what happens to accuracy where there is an unseen model in the blackbox under scrutiny. What are the generalization capabilities of Liteguard in open garden/sets scenarios? This must be discussed.

As one of the major motivations of the approach is a lower computational cost than MetaV for instance, and provided that both are compared in the experiments, I find it unfortunate not to have a plot or table measuring the gain (faster timings). To judge, we are left with a brief discussion on parameter complexity, that is interesting, but insufficient to support the claim in my opinion.

A proper related work section is missing, some papers are only briefly mentioned in the introduction. For instance, multiple papers on fingerprinting DNNs from recent conferences (e.g., AAAI 2025) are missing from the references.

In summary, I think this is an interesting lead, but that the experimental part in its current form is not enough to convey evidence and support the claims made at the beginning of the paper. The positioning w.r.t. the proliferation of prior work is also a problem.

**Questions:**

* how is is Litegard performing in an open garden setup?

* what about execution timings w.r.t. MetaV?

---

> ### Author Response · Authors · 2025-11-21
>
> ## Concern 1:
>
> Liteguard is not backed by any formal guarantee, the whole performance assessment is experiment-based. And as the experimental section relies on modest scenarios in practice (cifar 100...), it questions the performance to be expected in modern applications. In particular, it is clear that performance depends on the models present and compared in both sets; experiments only go up to 5 or 30 models and variants per set, which is a tiny number today (see e.g., "Fingerprinting Classiﬁers with Benign Inputs",2023 where 1046 variants are discriminated, for larger DNNs), facing the proliferation of architectures, proposals, and potential variants. This precludes a clear forecast of practical performances and generalization to other included models.
>
> **Response to concern 1:**
> Thank you for raising this concern. LiteGuard is a framework designed to improve model-set efficiency for task-agnostic fingerprinting. Similar to prior works—such as the task-agnostic MetaV and the task-specific IPGuard and GNNFinger—our study primarily relies on extensive empirical evaluations to demonstrate LiteGuard’s effectiveness in practical deployment settings. In addition to these comprehensive experiments, we also provide detailed justifications for our design choices, including the motivation for introducing a checkpoint-based model-set augmentation strategy and a local verifier architecture. These explanations help clarify the rationale behind LiteGuard and enable readers to better understand the framework.
>
> Regarding the model numbers, we would like to clarify that the “5–30 models’’ shown in Fig. 3 refers only to the number of trained models used to generate fingerprints—that is, the cost of constructing the model set. This does not represent the number of models used for testing. In fact, our testing model sets are larger than those used in prior work: the pirated and independently trained model sets used for evaluation in our experiments contain 91 and 100 models, respectively. By comparison, MetaV is evaluated on 70 pirated and 70 independent models, and ADV-TRA is evaluated on 50 pirated and 50 independent models. Importantly, all of these models are independently trained, and the consistent positive results observed across them demonstrate the robustness and effectiveness of LiteGuard. We believe this provides sufficient empirical evidence to support our claims.
>
> ---
>
> ## Concern 2:
>
> Lacks consideration of unseen models: this is not very clear in the paper, but it seems that all experiments consider a "closed garden" setup, where only models at hand are considered (ie models that have been part of the training for discrimination). In practice, and as in multiple papers in the SoA, the question is what happens to accuracy where there is an unseen model in the blackbox under scrutiny. What are the generalization capabilities of Liteguard in open garden/sets scenarios? This must be discussed.
>
> **Response to concern 2:**
> We would like to clarify that LiteGuard is not evaluated under a closed-garden setup. All models used for testing are entirely unseen—none of them is involved in the fingerprint-generation stage. The training model sets used to construct fingerprints are completely disjoint from the testing model sets.
>
> Accordingly, all reported AUC results already reflect generalization to new, previously unseen models, which corresponds precisely to the “open-garden” scenario described by the reviewer. This evaluation protocol is consistent with MetaV and other task-agnostic fingerprinting frameworks. By maintaining a strict separation between the training and testing model sets, LiteGuard’s generalization is assessed solely on models it has never encountered, accurately reflecting the practical requirements of open-set verification.
>
> ---

---

> ### Author Response · Authors · 2025-11-21
>
> ## Concern 3:
>
> As one of the major motivations of the approach is a lower computational cost than MetaV for instance, and provided that both are compared in the experiments, I find it unfortunate not to have a plot or table measuring the gain (faster timings). To judge, we are left with a brief discussion on parameter complexity, that is interesting, but insufficient to support the claim in my opinion.
>
> **Response to concern 3:**
> Thank you for raising this concern. The execution time of task-agnostic fingerprinting comes from two sources: the query-time overhead and the training time. For the query-time overhead, same as MetaV, LiteGuard also requires exactly one forward pass per fingerprint, and its verification consists of a simple aggregation of local-verifier outputs, leading to negligible computational cost. Instead, the training time is the dominant execution cost in task-agnostic fingerprinting, as pirated and independently-trained models need to be trained for constructing the two model sets.
>
> Therefore, LiteGuard’s computing efficiency gain comes from substantially reducing the number of trained models demanded. Fig. 3 illustrates the computational saving achieved by LiteGuard. Specifically, on the QM9 task, LiteGuard attains an AUC of 0.858 using only 2 trained models per set, already surpassing MetaV’s AUC of 0.849 obtained with 10 trained models—a 5× reduction in model-set training cost. With 5 trained models, LiteGuard reaches an AUC of 0.944, closely matching MetaV’s performance at 30 models, corresponding to a 6× reduction in training cost. A similar trend appears on the CASP regression task: LiteGuard with just 1 trained model exceeds MetaV’s performance at 10 models (a 10× reduction), and with 5 models matches MetaV at 30 models (a 6× reduction).
>
> These results demonstrate that LiteGuard achieves MetaV-level performance while using substantially fewer trained models—the primary contributor to computation cost—thereby delivering clear and significant efficiency gains even without additional timing curves. We hope our clarification can help address your concern.
>
> ---
>
> ## Concern 4:
>
> A proper related work section is missing, some papers are only briefly mentioned in the introduction. For instance, multiple papers on fingerprinting DNNs from recent conferences (e.g., AAAI 2025) are missing from the references.
>
> **Response to concern 4:**
> Thanks for pointing this out. We will add a related work section with more recent related papers in the revised version.
>
> ---
>
> ## Question 1:
>
> How is Litegard performing in an open garden setup?
>
> **Response to question 1:**
> We would like to clarify that LiteGuard has already evaluated in an open-garden setup. All models used for testing are entirely unseen—none of them is involved in the fingerprint-generation stage. The training model sets used to construct fingerprints are completely disjoint from the testing model sets.
>
> Accordingly, all reported AUC results already reflect generalization to new, previously unseen models, which corresponds precisely to the “open-garden” scenario described by the reviewer. This evaluation protocol is consistent with MetaV and other task-agnostic fingerprinting frameworks. By maintaining a strict separation between the training and testing model sets, LiteGuard’s generalization is assessed solely on models it has never encountered, accurately reflecting the practical requirements of open-set verification.

---

> ### Author Response · Authors · 2025-11-21
>
> ---
>
> ## Question 2:
>
> What about execution timings w.r.t. MetaV?
>
> **Response to question 2:**
> The execution time of task-agnostic fingerprinting comes from two sources: the query-time overhead and the training time. For the query-time overhead, same as MetaV, LiteGuard also requires exactly one forward pass per fingerprint, and its verification consists of a simple aggregation of local-verifier outputs, leading to negligible computational cost. Instead, the training time is the dominant execution cost in task-agnostic fingerprinting, as pirated and independently-trained models need to be trained for constructing the two model sets.
>
> Therefore, LiteGuard’s computing efficiency gain comes from substantially reducing the number of trained models demanded. Fig. 3 illustrates the computational saving achieved by LiteGuard. Specifically, on the QM9 task, LiteGuard attains an AUC of 0.858 using only 2 trained models per set, already surpassing MetaV’s AUC of 0.849 obtained with 10 trained models—a 5× reduction in model-set training cost. With 5 trained models, LiteGuard reaches an AUC of 0.944, closely matching MetaV’s performance at 30 models, corresponding to a 6× reduction in training cost. A similar trend appears on the CASP regression task: LiteGuard with just 1 trained model exceeds MetaV’s performance at 10 models (a 10× reduction), and with 5 models matches MetaV at 30 models (a 6× reduction).
>
> Thus, LiteGuard delivers concrete and substantial execution-time savings through dramatically reduced model-set training requirements, while keeping query-time cost identical to MetaV.

---

> ### Comment · Reviewer_GCH5 · 2025-11-25
> **reponse 1**
>
> I acknowledge authors' responses.
>
> Concern 1
>
> I do not buy this 'empirical only' is good enough argument. To make a
> parallel with the watermarking of models, initial scheme where
> empirical only; now the most recent are backup by guarantees, and this
> makes their contribution significantly stronger.
>
> Concern 2
>
> Close-garden vs "The training model sets used to construct
> fingerprints are completely disjoint from the testing model sets":
> there is a misunderstanding I believe, closed set means that the
> tested models have been leveraged for the fingerprint generation, here
> I think in the way you train with a protected model and an independent
> model.  In other words, what happens if after you did all your
> training and building stages, I test with say novel model architecture
> ABC, not seen anywhere in that previous phase, in one form or another
> ? This would be an open garden setup.
>
> In current proposal, all model architectures are placed in a pool for
> building fingerprints (model set construction), and at test time, some
> other models derived from the same architectures are leveraged for
> measurements.
>
> Concern 3
>
> Thanks for rephrasing, but I was mentioning timings, not relative training costs.
>
> Concern 4
>
> Waiting to witness this.

---

> > ### Author Response · Authors · 2025-12-02
> >
> > ## Concern 1
> >
> > I do not buy this 'empirical only' is good enough argument. To make a parallel with the watermarking of models, initial scheme where empirical only; now the most recent are backup by guarantees, and this makes their contribution significantly stronger.
> >
> > **Response 1:**
> > We appreciate the reviewer's use of watermarking to highlight the importance of theoretical guarantees. However, fingerprinting differs fundamentally from watermarking. In model watermarking, the embedded watermark signal is predefined, and the defender controls both the embedding mechanism and the training objective, which makes formal guarantees more accessible and has enabled recent theoretical advances. In contrast, fingerprinting seeks to identify special samples that serve as fingerprints without modifying the training of the protected model. Our proposed LiteGuard—like existing task-agnostic approaches such as TAFA and MetaV—derives fingerprints through a dedicated training process. As with training general AI models, providing theoretical guarantees for this fingerprint-training procedure is highly challenging.
> >
> > Instead, we offer thorough justifications for our design choices, including the motivations behind our checkpoint-based model-set augmentation strategy and local verifier architecture. Looking forward, rather than seeking theoretical guarantees, improving the explainability of fingerprint training, same as a lot of research work on enhancing the explainability of AI models, may be a promising research direction for advancing task-agnostic fingerprinting methods.

---

> > ### Author Response · Authors · 2025-12-02
> >
> > ## Concern 2
> > Closed set means that the tested models have been leveraged for the fingerprint generation.
> > In current proposal, all model architectures are placed in a pool for building fingerprints (model set construction), and at test time, some other models derived from the same architectures are leveraged for measurements.
> >
> > **Response 2:**
> > The reviewer mentioned “In current proposal, all model architectures are placed in a pool for building fingerprints (model set construction), and at test time, some other models derived from the same architectures are leveraged for measurements.” We believe there is a misunderstanding, though the model set construction part describes how to construct the model sets used in the fingerprint generation phase and the model sets for the testing phase. We would like to further clarify this more explicitly as follows.
> >
> > First, the model sets used for fingerprint generation contain only two architectures: one architecture for the model piracy set, which includes the protected model and its checkpoints, and another architecture for the independence set, which includes a single independently-trained model. Therefore, fingerprint generation does not involve “all architectures in the pool”; it uses exactly one architecture per set.
> >
> > Second, in contrast, the testing model sets—used solely to evaluate performance—are entirely disjoint from the fingerprint-generation model sets. The independent model set in the testing phase contains 7-10 different architectures depending on the task. Only one of these architectures (with a different random seed) overlaps with those used in the fingerprint-generation phase; all remaining architectures are completely unseen during fingerprint construction. Likewise, the pirated model set consists of six types of attacked variants of the protected model. None of these attacked models are involved in fingerprint generation, and they are therefore also unseen.
> >
> > In addition, we would like to clarify the applicability range of LiteGuard under broader “unknown model” scenarios that may arise in open-world settings.
> >
> > **AUCs achieved by LiteGuard under two independent model settings.**
> >
> > | Settings          | CNN/CIFAR-100 | MLP/CASP      | AE/CH         | GNN/QM9       | RNN/Weather   |
> > | ----------------- | ------------- | ------------- | ------------- | ------------- | ------------- |
> > | Original          | 0.936 ± 0.007 | 0.902 ± 0.007 | 0.977 ± 0.001 | 0.803 ± 0.003 | 0.971 ± 0.004 |
> > | Same archs&params | 0.931 ± 0.006 | 0.896 ± 0.007 | 0.967 ± 0.002 | 0.789 ± 0.004 | 0.944 ± 0.006 |
> >
> > First, we consider independent models trained on similar data, with the same architecture and hyperparameters as the protected model, but differing only in random seeds—a strictly challenging scenario. Such models already exist in our test set. To specifically quantify LiteGuard's discriminability under this condition, we further isolate them as the “Same archs&params” independent set, while the “Original setting” corresponds to the full independent set used in our main evaluation (which includes diverse architectures and also contains this subset). Across all five tasks, AnaFP exhibits only minor performance degradation under this stricter setting as reported in the table above. For example, the AUCs decrease from 0.936 to 0.931 on CNN/CIFAR-100, from 0.902 to 0.896 on MLP/CASP, from 0.977 to 0.967 on AE/CH, from 0.803 to 0.789 on GNN/QM9, and from 0.971 to 0.944 on RNN/Weather—demonstrating that LiteGuard retains strong discriminability even when independent models share the same architectures and training configurations (except random seed) with the protected model.
> >
> > Second, independent models that share the same architecture but differ in output from the protected model (e.g., identical output dimensionality but with incompatible output meanings) do not pose a problem for LiteGuard. Although the verifier can technically process their outputs, the functional behavior of such models is fundamentally unrelated to that of the protected model. As a result, the fingerprints—optimized to elicit characteristic responses from the protected model—are extremely unlikely to elicit similar behavior from such semantically mismatched models, resulting in low verification scores.
> >
> > Third, if the independent model does not even share the same output dimensionality as the protected model—the local verifier cannot operate on the model output due to the dimensional mismatch. In such cases, the model is immediately ruled out as a non-pirated model, since it falls outside the space where fingerprint responses can be meaningfully evaluated.
> >
> > Finally, pirated models produced through heavy transfer learning may deviate so substantially from the protected model that the functional geometry underpinning the fingerprints is no longer preserved—particularly when the data domain or output format changes. In such extreme cases, LiteGuard, like other fingerprinting schemes, may fail to detect ownership.

---

> > ### Author Response · Authors · 2025-12-02
> >
> > ---
> >
> > ## Concern 3
> >
> > Thanks for rephrasing, but I was mentioning timings, not relative training costs.
> >
> > **Response 3:**
> > LiteGuard’s computing efficiency gain comes from substantially reducing the number of trained models demanded. We analyze the computational saving achieved by LiteGuard reported in Fig. 3 with respect to the time cost in GPU hours. Specifically, on the QM9 task, LiteGuard attains an AUC of 0.858 using only 2 trained models per set (2 GPU hours), already surpassing MetaV’s AUC of 0.849 obtained with 10 trained models (10 GPU hours)—a 5× reduction in model-set training cost. With 5 trained models (5 GPU hours), LiteGuard reaches an AUC of 0.944, closely matching MetaV’s performance at 30 models (30 GPU hours), corresponding to a 6× reduction in training cost. A similar trend appears on the CASP regression task: LiteGuard with just 1 trained model (0.5 GPU hours) exceeds MetaV’s performance at 10 models (5 GPU hours)---a 10× reduction, and with 5 models (2.5 GPU hours) matches MetaV at 30 models (15 GPU hours)---a 6× reduction.
> >
> > These results demonstrate that LiteGuard achieves MetaV-level performance while using substantially fewer model training time costs—the primary contributor to computation cost—thereby delivering clear and significant efficiency gains.
> >
> > In practical scenarios where GPU resources are often scarce and expensive, the gain obtained by LiteGuard translates into substantial GPU-hour savings. In large-scale model management scenarios, when many models need to be fingerprinted, the cumulative reduction of 5×–10× per model compounds into a massive overall resource saving, underscoring the practical impact of LiteGuard’s efficiency advantage.
> >
> > We hope our clarification can help address your concern.
> >
> > ---
> >
> > ## Concern 4
> >
> > Waiting to witness this.
> >
> > **Response 4:**
> > In the revised version, we have added a dedicated Related Work section (Section G) in the Appendix and expanded the coverage of recent literature. In particular, we now include several works from very recent conferences, such as [1] and [2] from NeurIPS 2025, as well as [3] from AAAI 2025. The reviewer also mentioned that multiple fingerprinting papers appeared at AAAI 2025; among the four AAAI papers with “fingerprint” in the title, one [4] was already included in the first version of our submission, and another [3] has now been added. The remaining two papers focus on radio-frequency signal fingerprinting and decoding natural visual scenes from brain activity, respectively. While their titles contain the word “fingerprint,” they do not address DNN model fingerprinting and are therefore outside the scope of this paper.
> >
> > We hope this clarification helps, and we appreciate the reviewer’s suggestion, which has improved the completeness of the revised manuscript.
> >
> > [1] Zhuomeng Zhang, Fangqi Li, Hanyi Wang, and Shi-Lin Wang. *Boosting the uniqueness of neural networks fingerprints with informative triggers.* NeurIPS 2025.
> >
> > [2] Ling Tang, YueFeng Chen, Hui Xue, and Quanshi Zhang. *Towards the resistance of neural network fingerprinting to fine-tuning.* NeurIPS 2025.
> >
> > [3] Jianwei Fei, Yunshu Dai, Zhihua Xia, Fangjun Huang, and Jiantao Zhou. *Omnimark: efficient and scalable latent diffusion model fingerprinting.* AAAI 2025.
> >
> > [4] Augustin Godinot, Erwan Le Merrer, Camilla Penzo, Francois Taiani, and Gilles Tredan. *Queries, representation & detection: The next 100 model fingerprinting schemes.* AAAI 2025.

---

### Official Review · Reviewer_CS8j · 2025-10-31

**Soundness:** 3
**Presentation:** 4
**Contribution:** 3
**Rating:** 8
**Confidence:** 4

**Summary:**

This paper proposes LiteGuard, an efficient and task-agnostic model fingerprinting framework that enhances the generalization and computational efficiency of ownership verification compared to the state-of-the-art method MetaV. The authors identify that MetaV’s reliance on large, diverse training model sets and its globally entangled fingerprint–verifier architecture cause severe overfitting and high training cost. To address these issues, LiteGuard introduces two key innovations: (1) a checkpoint-based model set augmentation strategy that leverages intermediate training snapshots to enrich model diversity without additional training effort, and (2) a local verifier architecture that pairs each fingerprint with a lightweight, independent verifier to reduce parameter entanglement. Extensive experiments across five representative tasks (covering CNNs, MLPs, AEs, GNNs, and RNNs) demonstrate that LiteGuard achieves significantly higher AUC scores, superior robustness against six ownership-obfuscation techniques, and comparable verification accuracy, highlighting its practicality for efficient model IP protection.

**Strengths:**

1. **Elegant and Practical Design Innovations.** The checkpoint-based augmentation cleverly utilizes existing training artifacts to boost model diversity. The local verifier design directly reduces overfitting and decouples parameter dependencies. Both are low-cost, widely applicable strategies that can be generalized beyond fingerprinting.

2. **Strong Problem Motivation and Practical Relevance.** This paper addresses a real deployment bottleneck in model IP protection—MetaV’s dependence on large training model sets—and proposes a solution that is conceptually simple yet effective.

3. **Strong Empirical Results and Robustness.** LiteGuard shows large AUC gains over MetaV. The robustness experiments under six obfuscation types are particularly convincing.

4. **Excellent Writing and Clarity.** This paper is well-structured and clearly written.

**Weaknesses:**

1. **Limited Theoretical Analysis.** The paper’s central claim—improved generalization via reduced entanglement—is only empirically validated. There is no formal generalization or bias-variance analysis to support the intuition.

2. **Verifier Design.** The local verifier is a single linear layer; its capacity to handle high-dimensional nonlinear outputs is questionable. Runtime or query-efficiency trade-offs are not discussed.

**Questions:**

1. Could the authors provide a theoretical justification (even high-level) for why local verifiers improve generalization?

2. Would a small shared backbone with partially independent verifiers achieve better efficiency?

**Details Of Ethics Concerns:**

The paper’s focus is on ownership verification and IP protection, which supports ethical model usage and accountability. No privacy-violating or harmful applications are implied.

---

> ### Author Response · Authors · 2025-11-21
>
> ## Concern 1:
>
> Limited Theoretical Analysis. The paper’s central claim—improved generalization via reduced entanglement—is only empirically validated. There is no formal generalization or bias-variance analysis to support the intuition.
>
> **Response to concern 1:**
> Thank you for raising this concern. LiteGuard is a framework designed to improve model-set efficiency for task-agnostic fingerprinting. Similar to prior works—such as the task-agnostic MetaV and the task-specific IPGuard and GNNFinger—our study primarily relies on extensive empirical evaluations to demonstrate LiteGuard’s effectiveness in practical deployment settings. In addition to these comprehensive experiments, we also provide detailed justifications for our design choices, including the motivation for introducing a checkpoint-based model-set augmentation strategy and a local verifier architecture. These explanations help clarify the rationale behind LiteGuard and enable readers to better understand the framework.
>
> ---
>
> ## Concern 2:
>
> Verifier Design. The local verifier is a single linear layer; its capacity to handle high-dimensional nonlinear outputs is questionable. Runtime or query-efficiency trade-offs are not discussed.
>
> **Response to concern 2:**
> We would like to clarify that the local verifier is not a single linear layer. It consists of a linear projection followed by a nonlinear activation, which is sufficient for its intended role. The verifier is intentionally lightweight: its goal is not to approximate a complex nonlinear decision boundary, but to act as a binary discriminator over each fingerprint’s model response, leveraging the fact that the fingerprints themselves already encode task-agnostic discriminative structure.
>
> Regarding the runtime, the execution time of task-agnostic fingerprinting comes from two sources: the query-time overhead and the training time. For the query-time overhead, same as MetaV, LiteGuard also requires exactly one forward pass per fingerprint, and its verification consists of a simple aggregation of local-verifier outputs, leading to negligible computational cost. Instead, the training time is the dominant execution cost in task-agnostic fingerprinting, as pirated and independently-trained models need to be trained for constructing the two model sets.
>
> Therefore, LiteGuard’s computing efficiency gain comes from substantially reducing the number of trained models demanded. Fig. 3 illustrates the computational saving achieved by LiteGuard. Specifically, on the QM9 task, LiteGuard attains an AUC of 0.858 using only 2 trained models per set, already surpassing MetaV’s AUC of 0.849 obtained with 10 trained models—a 5× reduction in model-set training cost. With 5 trained models, LiteGuard reaches an AUC of 0.944, closely matching MetaV’s performance at 30 models, corresponding to a 6× reduction in training cost. A similar trend appears on the CASP regression task: LiteGuard with just 1 trained model exceeds MetaV’s performance at 10 models (a 10× reduction), and with 5 models matches MetaV at 30 models (a 6× reduction).
>
> Thus, LiteGuard delivers concrete and substantial execution-time savings through dramatically reduced model-set training requirements, while keeping query-time cost identical to MetaV. We hope the clarification has addressed your concern.
>
> ---
>
> ## Question 1:
>
> Could the authors provide a theoretical justification (even high-level) for why local verifiers improve generalization?
>
> **Response to Question 1:**
> A high-level justification is as follows. In MetaV, a single global verifier is jointly trained with all fingerprints, so a large set of parameters is optimized jointly. This entangled design tends to overfit when the number of available training models is small. In contrast, LiteGuard assigns each fingerprint an independent, lightweight verifier. Since each verifier is only jointly trained with one fingerprint, the total number of jointly trained parameters is greatly reduced, which can mitigate the overfitting issue and enhance generalization.
>
> ---

---

> ### Author Response · Authors · 2025-11-21
>
> ## Question 2:
>
> Would a small shared backbone with partially independent verifiers achieve better efficiency?
>
> **Response to Question 2:**
> Thank you for raising this insightful question. A shared-backbone design with partially independent verifiers is indeed a natural intermediate point between MetaV’s fully coupled global verifier and LiteGuard’s fully decoupled local verifiers.
>
> To explore this idea, we performed an additional experiment on the CNN/CIFAR-100 task. We keep the total number of fingerprints fixed at (N=100) and group them into sets of size (M). Each group shares a single verifier that jointly processes the outputs of its (M) fingerprints, resulting in (100/M) verifiers in total. Thus, larger (M) corresponds to more parameter sharing and greater cross-fingerprint coupling. We evaluate (M={1,2,5,10,50}), where (M=1) corresponds exactly to LiteGuard's original design.
>
> | (M) | AUC               |
> | --- | ----------------- |
> | 1   | (0.936 $\pm$ 0.007) |
> | 2   | (0.932 $\pm$ 0.007) |
> | 5   | (0.925 $\pm$ 0.011) |
> | 10  | (0.906 $\pm$ 0.014) |
> | 50  | (0.853 $\pm$ 0.023) |
>
> These results reveal a clear trend: as (M) increases and more fingerprints share the same verifier, AUC gradually decreases due to the reintroduction of jointly optimized parameters, which increases overfitting risk when only a small number of training models is available. However, when (M < 10), the degradation remains mild because the number of jointly trained parameters is still relatively small.
>
> More broadly, this hybrid design provides a **general framework** that unifies MetaV (full coupling) and LiteGuard (full decoupling) as two endpoints. We speculate that different conditions (e.g., different training model-set sizes) may favor different levels of coupling. Identifying the optimal grouping size (M) under various conditions is an interesting direction for future work.

---

> ### Comment · Reviewer_CS8j · 2025-11-27
> **Response from Reviewer CS8j**
>
> I acknowledge authors' responses.
>
> About Concern 1: I still insist that there should be some theoretical analysis about the generalization via reduced entanglement and checkpoint-based model set augmentation strategy, as this is the main difference between this work and existing works. And the readers might be curious about where the gains are from. Thus, there should be at least some reasonable justifications for this, rather than "empirical only".
>
> About Concern 2: The training time for this fingerprinting should be much longer than that of existing methods. Would this be a limitation?
>
> About other questions: Thanks authors for your responses.

---

> > ### Author Response · Authors · 2025-12-02
> >
> > ## Concern 1:
> >
> > I still insist that there should be some theoretical analysis about the generalization via reduced entanglement and checkpoint-based model set augmentation strategy, as this is the main difference between this work and existing works. And the readers might be curious about where the gains are from. Thus, there should be at least some reasonable justifications for this, rather than "empirical only".
> >
> > **Response 1:**
> > Fingerprinting seeks to identify special samples that serve as fingerprints without modifying the training of the protected model. Our proposed LiteGuard—like existing task-agnostic approaches such as TAFA and MetaV—derives fingerprints through a dedicated training process. As with training general AI models, providing theoretical guarantees for this fingerprint-training procedure is highly challenging. Instead, we offer the following high-level justification for why LiteGuard’s two core design choices are effective.
> >
> > First, checkpoint-based model-set augmentation reduces variance by enriching behavioral diversity. Each DNN checkpoint captured during training represents distinct intermediate decision behaviors. Incorporating these checkpoints—without training additional models—creates a more diverse sampling of model behaviors in both piracy and independence sets. This expanded diversity prevents the fingerprint–verifier pairs from overfitting to a small set of models, effectively lowering the variance. This is analogous to data augmentation in standard model learning: more behavioral “samples” allow a more stable discrimination boundary.
> >
> > Second, the local verifier architecture mitigates parameter entanglement and controls complexity. MetaV jointly trains all fingerprints with a global neural verifier. This causes the verifier’s input dimension and parameter count to scale with the number of fingerprints, making the joint training highly entangled and prone to severe overfitting when model sets are small. In contrast, LiteGuard pairs each fingerprint with an independent lightweight local verifier. This achieves two effects: 1) It decouples the optimization across fingerprints, drastically reducing the number of jointly trained parameters; and 2) It guarantees parameter complexity remains constant regardless of the fingerprint set size.
> >
> > ---
> >
> > ## Concern 2
> >
> > The training time for this fingerprinting should be much longer than that of existing methods. Would this be a limitation?
> >
> > **Response 2:**
> > If the reviewer refers to “training time” as the time of directly training fingerprint–verifier pairs, we note that this cost is very small. Each fingerprint–verifier pair contains only a few number of parameters, and during training we adopt a batch size K=1, meaning that every iteration it samples only one model from each set, requiring just a single forward--backward pass on two models. Besides, the fingerprint-verifier pairs are totally independent, making their training parallelizable. As a result, training fingerprint-verifier pairs is highly efficient. For example, on the CNN/CIFAR-100 task, training 100 fingerprint–verifier pairs takes only about 3 minutes, which is comparable to existing methods and is an acceptable cost.
> >
> > If the reviewer instead refers to “training time” as the time of training the model sets (i.e., piracy and independence sets), LiteGuard is much efficient than existing task-agnostic methods, such as MetaV and TAFA. This is because LiteGuard targets to the reduction of training time for constructing the model sets by adopting the checkpoint-based augmentation strategy and the local verifier design, allowing us to train on far fewer models.
> >
> > If the reviewer compares the time with existing task-specific fingerprinting methods, we notice that some of them do not require training any model sets, incurring no model training time. However, these methods cannot operate in a task-agnostic setting. In contrast, LiteGuard targets the more challenging task-agnostic regime, where model-set construction is unavoidable.

---

### Official Review · Reviewer_588g · 2025-11-01

**Soundness:** 3
**Presentation:** 3
**Contribution:** 3
**Rating:** 6
**Confidence:** 4

**Summary:**

This paper proposed LITEGUARD, a Task-agnostic model fingerprinting model fingerprinting framework with enhanced generalization. It features two key innovations, (1) using intermediate model check-points from training to enrich the model set, and (2) using a local verifier architecture that all fingerprints are optimized independently. Empirical results are provided to show that the proposed method achieves a better performance as well as a lower computation cost.

**Strengths:**

1. This paper is well structured and easy to follow.

2. The empirical results are strong and the experiment design is fair. It shows a large margin of improvement given by the proposed method compared to the baselines.

3. Very rich experimental results. It considered both scenarios with and without obfuscation methods.

4. Ablation studies are also well designed and convincing.

**Weaknesses:**

1. The novelty of this paper is relatively weak. Independently optimizing different fingerprints is also used in other previous works. Although these works are not considered task-agnostic, it would be better to clarify how this work differs from them.

2, Font size in Fig. 4 is too small to read.

**Questions:**

1. In the experiments, is the number of models (including different model snapshots) the same between the proposed method and baselines?

2. In the complexity discussion in section 4 "LiteGuard: Decoupled Architecture", shouldn't it have a factor N in the formula?

---

> ### Author Response · Authors · 2025-11-21
>
> ## Concern 1:
>
> The novelty of this paper is relatively weak. Independently optimizing different fingerprints is also used in other previous works. Although these works are not considered task-agnostic, it would be better to clarify how this work differs from them.
>
> **Response to concern 1:**
> Thanks for raising this concern. Although like almost all previous task-specific methods, LiteGuard also independently optimizes fingerprints, our work differs from them in several essential ways. First, the design choice is different: previous methods only optimize fingerprints, whereas we jointly optimize fingerprint–verifier pairs. Second, the capability is different: those methods only work for specific tasks (typically the classification task), whereas ours is widely applicable to various tasks. Third, the motivation is different: prior works happen to optimize fingerprints independently, while our design intentionally introduces independence to reduce parameter complexity and improve generalization in task-agnostic settings.
>
> This new design of the local verifier architecture enables the reduction of parameter entanglement and the mitigation of overfitting. Moreover, our another innovation is to propose a check-point-based model set augmentation strategy to enrich model diversity by leveraging intermediate model snapshots captured during the training of each pirated and independently-trained model. This novel design can reduce computational costs significantly. We hope our clarification has already addressed well your concern.
>
> ---
>
> ## Concern 2:
>
> Font size in Fig. 4 is too small to read.
>
> **Response to concern 2:**
> Thanks for pointing this out. We will fix it in the revised version and carefully proofread the whole paper.
>
> ---
>
> ## Question 1:
>
> In the experiments, is the number of models (including different model snapshots) the same between the proposed method and baselines?
>
> **Response to question 1:**
> Yes, the number of trained models used by our method and the baselines is the same, meaning the training cost for producing these models is identical. The difference is that LiteGuard additionally includes the checkpoints generated during the training of these models to increase the diversity of the model sets. This improves generalization without introducing any extra model-training cost. It is noted that task-specific baselines IPGuard and ADVTRA do not utilize any model sets during fingerprint generation.
>
> ---
>
> ## Question 2:
>
> In the complexity discussion in section 4 "LiteGuard: Decoupled Architecture", shouldn't it have a factor N in the formula?
>
> **Response to question 2:**
> In Section 4, we analyze the number of jointly trained parameters because overfitting risk is driven by parameters that are optimized together. LiteGuard trains each fingerprint–verifier pair independently, so these parameters are not jointly optimized across the N pairs. Therefore, the complexity does not scale with N. In contrast, MetaV jointly trains all N fingerprints with a single global verifier, which is why its complexity includes an explicit N factor.

---

### Official Review · Reviewer_QEjw · 2025-11-03

**Soundness:** 3
**Presentation:** 4
**Contribution:** 3
**Rating:** 6
**Confidence:** 4

**Summary:**

This paper introduces LiteGuard, an efficient and task-agnostic framework for model fingerprinting designed to enhance generalization and reduce computational overhead in ownership verification of deep neural networks. Building upon the limitations of prior work such as MetaV—which relies on extensive model collections and joint training of entangled fingerprints and verifiers—LiteGuard contributes two core innovations: a checkpoint-based model set augmentation strategy that increases model diversity by reusing intermediate training checkpoints, and a local verifier architecture that pairs each fingerprint with an independent, lightweight verifier to mitigate overfitting. Extensive experiments across five representative tasks—spanning classification, regression, generation, and graph modeling—demonstrate that LiteGuard achieves superior generalization and computational efficiency, while maintaining robustness against a wide range of ownership obfuscation attacks. This paper offers a well-motivated and technically grounded contribution that effectively addresses a genuine limitation of existing task-agnostic fingerprinting methods. While it represents a solid and practically meaningful advance, further analytical depth would strengthen its impact.

**Strengths:**

1. Well-identified practical bottleneck (generalization vs. efficiency trade-off).

The authors clearly articulate that the core limitation of existing task-agnostic fingerprinting, especially MetaV, lies in its dependence on large and diverse model collections to achieve generalization. This observation grounds the work in a genuine real-world constraint: publicly available models are scarce, and constructing them is resource-intensive. The authors convincingly motivate the need for a lightweight yet generalizable alternative, establishing a coherent link between practical constraints and methodological innovation. This framing situates LiteGuard as a realistic and deployable solution to a widely recognized challenge in model IP protection.

2. Checkpoint-based model set augmentation strategy.

The checkpoint-based augmentation mechanism is one of the paper’s most compelling contributions. By systematically incorporating intermediate model snapshots captured during training, LiteGuard effectively transforms existing computational by-products into valuable training diversity (Sec. 3.2; L162–L215). The checkpoint selection scheme—starting from a mid-training epoch and uniformly sampling at fixed intervals—balances representational variety and redundancy. The ablation results in Figure 4 empirically confirm that this augmentation consistently improves AUCs across tasks, even when applied to MetaV, illustrating its general applicability. Conceptually, this design elegantly aligns with the paper’s goal of efficiency without sacrificing robustness, and it represents a practically impactful contribution to fingerprint generation under constrained resources.

3. Modular local verifier architecture with analytical justification.

LiteGuard’s second key innovation is its decoupled verifier structure, which addresses the overfitting and scalability issues inherent in MetaV’s global verifier design. Each fingerprint is paired with an independent, lightweight verifier trained only on its own outputs (Sec. 3.3; L213–L217), dramatically reducing parameter entanglement. The formal complexity analysis in Sec. 4 shows that this approach lowers parameter scaling from O(N(F+O⋅H)) to O(F+O), thereby bounding overfitting risk. This theoretical rationale is reinforced experimentally: Figure 5 shows that substituting LiteGuard’s verifier into MetaV significantly boosts performance, while removing it leads to a marked decline. The synergy of analytical reasoning and empirical validation gives this architectural refinement both conceptual soundness and demonstrable impact.

4. Comprehensive and multi-domain empirical validation.

The experimental design is broad and methodologically rigorous. LiteGuard is evaluated across five tasks—classification, regression, graph prediction, tabular data generation, and time-series modeling—each involving distinct architectures and datasets (Sec. 5; Table 2). The results reveal consistent AUC improvements over both task-specific and task-agnostic baselines, with gains ranging from 9% to 35%. Moreover, Figure 3 demonstrates that LiteGuard achieves comparable performance to MetaV while using up to 80 % fewer trained models, confirming its superior efficiency. Table 3 further shows that LiteGuard maintains robustness under diverse ownership obfuscation techniques, outperforming MetaV across nearly all categories. The breadth and consistency of these results give the work strong empirical credibility.

**Weaknesses:**

1. Limited theoretical understanding of generalization improvement.

Section 4 provides a parameter-count argument suggesting that reduced entanglement mitigates overfitting, but it stops short of a quantitative or theoretical treatment. There is no analysis of how checkpoint diversity or modularity affects the variance or bias of fingerprint representations, nor are there experiments correlating parameter scale with test AUC. As a result, the claim of “enhanced generalization” remains empirically supported but theoretically underdeveloped, limiting the scientific depth of the contribution.

2. Lack of theoretical justification for why checkpoint augmentation and local verifiers work.

The paper convincingly shows that both the checkpoint-based augmentation and the local verifier design improve performance, but it is unclear whether checkpoints truly capture orthogonal decision behaviors or merely introduce correlated noise, and whether independent verifiers learn complementary discriminative subspaces or simply reduce parameter coupling.

3. Incomplete analysis of computational efficiency and broader implications.

While the efficiency advantage is repeatedly emphasized, the paper evaluates it solely through AUC comparisons at different model counts (Fig. 3) rather than concrete computational metrics such as training time, GPU hours, or memory usage. This omission makes it difficult to quantify the real-world resource savings claimed in the abstract. Moreover, the conclusion section does not discuss potential security or ethical implications—such as false positives in ownership verification or adversarial adaptation to the local verifier design—leaving the practical boundaries of LiteGuard’s applicability somewhat underexplored.

**Questions:**

1. Could the authors provide quantitative evidence of computational efficiency?

2. Would it be possible to characterize the relationship between checkpoint diversity, parameter decoupling, and generalization empirically or theoretically?

3. Could the authors complement Tables 2 and 3 with some statistical significance tests or confidence intervals to ensure the reported improvements are not due to stochastic variation?

4. Would a comparative summary table outlining architectural dependencies, training assumptions, and verifier coupling help clarify the theoretical boundary of LiteGuard’s contributions relative to existing methods?

---

> ### Author Response · Authors · 2025-11-21
>
> ## Concern 1:
>
> Limited theoretical understanding of generalization improvement.
> Section 4 provides a parameter-count argument suggesting that reduced entanglement mitigates overfitting, but it stops short of a quantitative or theoretical treatment. There is no analysis of how checkpoint diversity or modularity affects the variance or bias of fingerprint representations, nor are there experiments correlating parameter scale with test AUC. As a result, the claim of “enhanced generalization” remains empirically supported but theoretically underdeveloped, limiting the scientific depth of the contribution.
>
> **Response to concern 1:**
> We appreciate the reviewer’s desire for a deeper theoretical treatment. LiteGuard provides a framework for improving the practicality and generalization of task-agnostic fingerprinting, a setting that spans arbitrary tasks and model families. Same as prior task-agnostic approaches like MetaV, providing a fully-formal generalization analysis is challenging. Nonetheless, our design is supported by both conceptual justification and empirical evidence.
>
> Regarding how checkpoint diversity and modularity influence bias and variance, we clarify the relationship as follows. Checkpoint-based augmentation increases the diversity of model behaviors that each fingerprint–verifier pair encounters during training, which makes the learned decision rule less sensitive to any single model and therefore reduces variance. Modularity further lowers variance by reducing the number of jointly trained parameters, which is crucial when the number of available training models is limited. Bias is not the limiting factor in this setting—the fingerprint–verifier pair already offers sufficient expressive power for the binary discrimination task. This is supported by the experimental results on GNN/QM9 in the following table, where deeper verifier architectures do not yield performance gains. Instead, they reduce AUC, indicating that the added capacity increases the number of jointly optimized parameters and ultimately harms generalization.
>
> **Impact of verifier depth on GNN/QM9 AUC.**
>
> | Verifier Depth | AUC (QM9)         |
> | -------------- | ----------------- |
> | 1 layer        | $0.803 \pm 0.003$ |
> | 2 layers       | $0.792 \pm 0.007$ |
> | 3 layers       | $0.766 \pm 0.021$ |
> | 4 layers       | $0.725 \pm 0.078$ |
>
> ---
>
> ## Concern 2:
>
> Lack of theoretical justification for why checkpoint augmentation and local verifiers work.
> The paper convincingly shows that both the checkpoint-based augmentation and the local verifier design improve performance, but it is unclear whether checkpoints truly capture orthogonal decision behaviors or merely introduce correlated noise, and whether independent verifiers learn complementary discriminative subspaces or simply reduce parameter coupling.
>
> **Response to concern 2:**
> We thank the reviewer for the insightful question.
>
> LiteGuard is a framework designed to improve model-set efficiency for task-agnostic fingerprinting. Similar to prior works—such as the task-agnostic MetaV and the task-specific IPGuard and GNNFinger—our study primarily relies on extensive empirical evaluations to demonstrate LiteGuard’s effectiveness in practical deployment settings. In addition to these comprehensive experiments, we also provide detailed justifications for our design choices, including the motivation for introducing a checkpoint-based model-set augmentation strategy and a local verifier architecture. These explanations help clarify the rationale behind LiteGuard and enable readers to better understand the framework.
>
> Checkpoints do not behave as correlated noise. As shown in Fig. 4, performance depends strongly on checkpoint stage: early checkpoints degrade performance (near-random behavior), very late checkpoints add little diversity, while mid-stage checkpoints significantly improve AUC. This non-monotonic pattern indicates that checkpoints encode distinct decision behaviors.
>
> Local verifiers indeed reduce parameter coupling, which is the primary mechanism for improving generalization.
> Moreover, if decoupling were the only effect, we would expect the performance to saturate once a small number of fingerprint–verifier pairs is used. However, we observe from Figure 6 that LiteGuard’s AUC keeps improving as the number of pairs increases (from 10 up to 100), with decreasing variance. This indicates that different pairs do not learn identical signals but provide additional non-redundant discriminative cues beyond mere parameter decoupling.

---

> ### Author Response · Authors · 2025-11-21
>
> ---
>
> ## Concern 3:
>
> Incomplete analysis of computational efficiency and broader implications.
> While the efficiency advantage is repeatedly emphasized, the paper evaluates it solely through AUC comparisons at different model counts (Fig. 3) rather than concrete computational metrics such as training time, GPU hours, or memory usage. This omission makes it difficult to quantify the real-world resource savings claimed in the abstract. Moreover, the conclusion section does not discuss potential security or ethical implications—such as false positives in ownership verification or adversarial adaptation to the local verifier design—leaving the practical boundaries of LiteGuard’s applicability somewhat underexplored.
>
> **Response to concern 3:**
> Thank you for raising this concern. The execution time of task-agnostic fingerprinting comes from two sources: the query-time overhead and the training time. For the query-time overhead, same as MetaV, LiteGuard also requires exactly one forward pass per fingerprint, and its verification consists of a simple aggregation of local-verifier outputs, leading to negligible computational cost. Instead, the training time is the dominant execution cost in task-agnostic fingerprinting, as pirated and independently-trained models need to be trained for constructing the two model sets.
>
> Therefore, LiteGuard’s computing efficiency gain comes from substantially reducing the number of trained models demanded. Fig. 3 illustrates the computational saving achieved by LiteGuard. Specifically, on the QM9 task, LiteGuard attains an AUC of 0.858 using only 2 trained models per set, already surpassing MetaV’s AUC of 0.849 obtained with 10 trained models—a 5× reduction in model-set training cost. With 5 trained models, LiteGuard reaches an AUC of 0.944, closely matching MetaV’s performance at 30 models, corresponding to a 6× reduction in training cost. A similar trend appears on the CASP regression task: LiteGuard with just 1 trained model exceeds MetaV’s performance at 10 models (a 10× reduction), and with 5 models matches MetaV at 30 models (a 6× reduction). These results demonstrate that LiteGuard achieves MetaV-level performance while using substantially fewer trained models—the primary contributor to computation cost—thereby delivering clear and significant efficiency gains even without additional timing curves. We hope our clarification can help address your concern.
>
> Regarding broader implications, we will add a brief discussion to clarify practical boundaries. Regarding false positives, we agree this is an inherent issue for all ownership-verification methods, as any fingerprinting scheme may occasionally classify an independently trained model as pirated. LiteGuard does not explicitly target false-positive elimination, but this risk can be reduced by setting a higher verification threshold. Moreover, we envision fingerprinting-based verification as an initial screening tool that provides an evidence for subsequent, more rigorous verification procedures that will rule out the occasional false positives. Regarding adversarial adaptation to the local verifier, LiteGuard limits such adaptation for the following two reasons. First, the verifier is kept confidential to attackers. Second, each verifier is independent. Thus, an attacker would need to manipulate the outputs of many verifiers simultaneously to meaningfully change the final aggregated score, making such adaptation considerably more difficult.

---

> ### Author Response · Authors · 2025-11-21
>
> ## Question 1:
>
> Could the authors provide quantitative evidence of computational efficiency?
>
> **Response to question 1:**
> Yes. In our setting, “computational efficiency” refers to the computing cost of training the model sets required by task-agnostic fingerprinting methods. The execution time of task-agnostic fingerprinting comes from two sources: the query-time overhead and the training time. For the query-time overhead, same as MetaV, LiteGuard also requires exactly one forward pass per fingerprint, and its verification consists of a simple aggregation of local-verifier outputs, leading to negligible computational cost. Instead, the training time is the dominant execution cost in task-agnostic fingerprinting, as pirated and independently-trained models need to be trained for constructing the two model sets.
>
> Therefore, LiteGuard’s computing efficiency gain comes from substantially reducing the number of trained models demanded. Fig. 3 illustrates the computational saving achieved by LiteGuard. Specifically, on the QM9 task, LiteGuard attains an AUC of 0.858 using only 2 trained models per set, already surpassing MetaV’s AUC of 0.849 obtained with 10 trained models—a 5× reduction in model-set training cost. With 5 trained models, LiteGuard reaches an AUC of 0.944, closely matching MetaV’s performance at 30 models, corresponding to a 6× reduction in training cost. A similar trend appears on the CASP regression task: LiteGuard with just 1 trained model exceeds MetaV’s performance at 10 models (a 10× reduction), and with 5 models matches MetaV at 30 models (a 6× reduction).
>
> These results demonstrate that LiteGuard provides comparable or stronger verification performance with dramatically fewer trained models, offering concrete and substantial computational savings.
>
> ---
>
> ## Question 2:
>
> Would it be possible to characterize the relationship between checkpoint diversity, parameter decoupling, and generalization empirically or theoretically?
>
> **Response to question 2:**
> We appreciate the reviewer’s question. Providing a full theoretical characterization of how checkpoint diversity and parameter decoupling affect generalization remains highly challenging. Nevertheless, LiteGuard already offers both conceptual justification and empirical evidence for this relationship. Our ablations (Figs. 4 and 5 in the original paper) show that (i) increasing checkpoint diversity improves discriminative performance, and (ii) reducing parameter entanglement through local verifiers yields better performance.

---

> ### Author Response · Authors · 2025-11-21
>
> ---
>
> ## Question 3:
>
> Could the authors complement Tables 2 and 3 with some statistical significance tests or confidence intervals to ensure the reported improvements are not due to stochastic variation?
>
> **Response to question 3:**
>
> **95% confidence intervals (CI) for AUCs by different approaches across five tasks (different model types and datasets).**
>
> | Method     | CNN/CIFAR-100  | MLP/CASP       | AE/CH          | GNN/QM9        | RNN/Weather    |
> | ---------- | -------------- | -------------- | -------------- | -------------- | -------------- |
> | UAP        | [0.696, 0.808] | --             | --             | --             | --             |
> | IPGuard    | [0.640, 0.776] | --             | --             | --             | --             |
> | ADVTRA     | [0.833, 0.857] | --             | --             | --             | --             |
> | GNNFingers | --             | --             | --             | [0.403, 0.813] | --             |
> | MetaV      | [0.652, 0.699] | [0.814, 0.834] | [0.747, 0.819] | [0.464, 0.732] | [0.827, 0.881] |
> | LiteGuard  | [0.927, 0.945] | [0.893, 0.911] | [0.976, 0.978] | [0.799, 0.807] | [0.966, 0.976] |
>
> **95% confidence intervals (CI) for AUCs under various ownership obfuscation techniques across various tasks.**
>
> | Dataset       | Method        | Pruning            | Finetuning         | KD                 | N-Finetune         | P-Finetune         | A-Finetune         |
> | ------------- | ------------- | ------------------ | ------------------ | ------------------ | ------------------ | ------------------ | ------------------ |
> | CNN/CIFAR-100 | UAP           | [0.647, 0.815]     | [0.899, 0.907]     | [0.614, 0.792]     | [0.575, 0.709]     | [0.690, 0.814]     | [0.698, 0.862]     |
> | CNN/CIFAR-100 | IPGuard       | [0.645, 0.839]     | [0.976, 1.004]     | [0.544, 0.618]     | [0.690, 0.750]     | [0.632, 0.674]     | [0.527, 0.595]     |
> | CNN/CIFAR-100 | ADVTRA        | [0.873, 0.901]     | [0.987, 0.999]     | [0.806, 0.826]     | [0.715, 0.737]     | [0.878, 0.916]     | [0.855, 0.883]     |
> | CNN/CIFAR-100 | MetaV         | [0.691, 0.765]     | [0.755, 0.797]     | [0.576, 0.630]     | [0.691, 0.723]     | [0.666, 0.696]     | [0.489, 0.533]     |
> | CNN/CIFAR-100 | **LiteGuard** | **[0.946, 0.964]** | **[0.994, 0.998]** | **[0.775, 0.797]** | **[0.980, 0.992]** | **[0.963, 0.987]** | **[0.975, 0.985]** |
> | MLP/CASP      | MetaV         | [0.991, 1.003]     | [0.976, 1.000]     | [0.473, 0.567]     | [0.968, 1.006]     | [0.610, 0.784]     | [0.816, 0.956]     |
> | MLP/CASP      | **LiteGuard** | **[0.965, 0.997]** | **[1.000, 1.000]** | **[0.529, 0.539]** | **[1.000, 1.000]** | **[0.981, 0.989]** | **[0.976, 0.980]** |
> | AE/CH         | MetaV         | [0.636, 0.674]     | [0.763, 0.889]     | [0.771, 0.825]     | [0.781, 0.843]     | [0.752, 0.806]     | [0.795, 0.807]     |
> | AE/CH         | **LiteGuard** | **[0.921, 0.933]** | **[0.992, 0.992]** | **[0.991, 0.991]** | **[0.990, 0.992]** | **[0.974, 0.976]** | **[0.991, 0.991]** |
> | GNN/QM9       | GNNFingers    | [0.550, 0.826]     | [0.307, 0.967]     | [0.376, 0.636]     | [0.346, 0.832]     | [0.466, 0.776]     | [0.585, 0.661]     |
> | GNN/QM9       | MetaV         | [0.615, 0.731]     | [0.400, 0.802]     | [0.525, 0.575]     | [0.376, 0.784]     | [0.559, 0.803]     | [0.574, 0.628]     |
> | GNN/QM9       | **LiteGuard** | **[0.983, 0.985]** | **[0.828, 0.840]** | **[0.639, 0.651]** | **[0.787, 0.795]** | **[0.807, 0.817]** | **[0.768, 0.772]** |
> | RNN/Weather   | MetaV         | [0.848, 0.892]     | [0.779, 0.853]     | [0.900, 0.944]     | [0.815, 0.855]     | [0.806, 0.864]     | [0.736, 0.788]     |
> | RNN/Weather   | **LiteGuard** | **[0.988, 0.988]** | **[0.955, 0.973]** | **[0.987, 0.995]** | **[0.956, 0.968]** | **[0.950, 0.964]** | **[0.945, 0.949]** |
>
> To confirm that LiteGuard’s improvements are not due to stochastic variation, we computed 95% confidence intervals (CI) for all AUC values in the above tables. Across all datasets and obfuscation settings, LiteGuard’s confidence intervals do not overlap with those of MetaV or other baselines. These results demonstrate that LiteGuard’s performance improvement is statistically significant.

---

> ### Author Response · Authors · 2025-11-21
>
> ---
>
> ## Question 4:
>
> Would a comparative summary table outlining architectural dependencies, training assumptions, and verifier coupling help clarify the theoretical boundary of LiteGuard’s contributions relative to existing methods?
>
> **Response to question 4:**
> Thanks for your constructive suggestion. We agree that such a comparative summary would help clarify the boundary of LiteGuard relative to prior work to highlight the contributions. Therefore, we provide the following table to summarize the comparison of fingerprint methods along three key axes, including task-agnostic, model-set training cost, and scalability. The table shows that only MetaV and LiteGuard are task-agnostic, i.e., applicable for different tasks. Compared with MetaV, LiteGuard requires much lower model-set training cost and stronger scalability. In particular, the decoupled verifier design makes LiteGuard highly scalable—new fingerprint-verifier pairs can be added without retraining existing ones, and verification can flexibly adapt to different query budgets by using a corresponding number of fingerprint-verifier pairs. In contrast, MetaV requires retraining all fingerprints and the verifier together whenever the fingerprint set changes. In the revised version, we will add the table in the appendix.
>
> **Comparison of fingerprinting methods along three key axes.**
>
> | Method           | Task-agnostic? | Model-set training cost | Scalability |
> | ---------------- | -------------- | ----------------------- | -------------------- |
> | IPGuard          | No             | --                      | High                  |
> | UAP              | No             | Low--Moderate           | High                  |
> | ADVTRA           | No             | --                      | High                 |
> | GNNFingers       | No             | High                    | Low                   |
> | MetaV            | Yes            | High                    | Low                  |
> | LiteGuard (ours) | Yes            | Low                     | High                  |

---

### Meta-Review · Area_Chair_4B39 · 2026-01-02

**Summary:**

This paper proposes LiteGuard, an efficient task-agnostic model fingerprinting framework that improves generalization under limited training-model availability. It introduces two key ideas: (i) checkpoint-based model-set augmentation that reuses intermediate training snapshots to increase model diversity without training additional models, and (ii) a modular local-verifier architecture that pairs each fingerprint with an independent lightweight verifier, reducing parameter entanglement and mitigating overfitting. Across five representative tasks spanning classification, regression, generation, and graph modeling, LiteGuard substantially outperforms the prior task-agnostic baseline MetaV in generalization and robustness, including under multiple ownership-obfuscation techniques, while requiring far fewer trained models to reach similar AUC levels. Reviewers generally find the paper well-motivated, clearly written, and empirically strong; the rebuttal further strengthens the submission by providing confidence intervals for statistical significance, clarifying the training/testing protocol as an “unseen-model” evaluation, and adding GPU-hour estimates and additional ablations on verifier coupling. Remaining concerns mainly relate to the limited depth of theoretical analysis and the precise boundaries of “open-world” generalization, but these do not appear to undermine the central empirical claims. Overall, the paper represents a practically meaningful advance over MetaV and should be competitive for acceptance.

**Reviewer Concerns:**

Reviewers’ positive feedback converges on the clarity of motivation and the effectiveness of the two design choices. The checkpoint-based augmentation is viewed as a practically impactful way to extract diversity “for free” from training artifacts, and the local-verifier architecture is supported by both a parameter-complexity argument and consistent empirical gains. The experiments are broad and well-controlled: LiteGuard is evaluated across five tasks and model families, compared against MetaV as the primary task-agnostic baseline, and additionally against relevant task-specific baselines where applicable; robustness is assessed under six ownership-obfuscation techniques. The rebuttal additionally supplies 95% confidence intervals and demonstrates non-overlapping CIs versus MetaV across tasks and obfuscation settings, supporting statistical significance, and provides further experiments on intermediate coupling designs that interpolate between LiteGuard and MetaV.


The main remaining concerns are: (i) theoretical depth: several reviewers request a more formal generalization/bias–variance analysis beyond parameter-count intuition; (ii) clarity on “open-world” deployment boundaries: one reviewer argues that while train/test model sets are disjoint, many test architectures overlap with those used during fingerprint construction, and the strongest notion of “open garden” would include entirely novel architectures or output formats; and (iii) computational reporting: initially the paper lacked explicit timing/GPU-hour measurements and relied on model-count proxies, though the rebuttal partially addresses this by relating model-count to GPU hours and clarifying that fingerprint–verifier training itself is lightweight and parallelizable.

**Reviewer Scores:**

Reviewer scores are predominantly above the acceptance threshold (including an 8/10 accept and two 6/10 weak accepts), with one reviewer at borderline reject primarily due to the lack of formal guarantees and a stricter interpretation of “open garden” evaluation. The rebuttal addresses multiple actionable concerns (statistical significance via confidence intervals, GPU-hour estimates, clarification of disjoint train/test model sets, and additional experiments on partially shared verifiers), which should stabilize or modestly increase scores for reviewers who were already positive. The remaining disagreement is largely philosophical (expecting guarantees in fingerprinting) and definitional (how strict “open-world” should be). Under a full discussion period, it is plausible that the borderline reviewer might remain unconvinced, but the overall consensus would still most likely converge to a weak accept/accept decision given the strength and consistency of the empirical gains and the clarified evaluation protocol.

---

### Decision · Program_Chairs · 2026-01-26

Accept (Poster)